# Synergizing Large Language Models and Task-specific Models for Time Series Anomaly Detection

## Abstract

In anomaly detection, methods based on large language models (LLMs) can incorporate expert knowledge by reading professional document, while task-specific small models excel at extracting normal data patterns and detecting value fluctuations from training data of target applications. Inspired by the human nervous system, where the brain stores expert knowledge and the peripheral nervous system and spinal cord handle specific tasks like withdrawal and knee-jerk reflexes, we propose CoLLaTe, a framework designed to facilitate collaboration between LLMs and task-specific models, leveraging the strengths of both models for anomaly detection. In particular, we first formulate the collaboration process and identify two key challenges in the collaboration: (1) the misalignment between the expression domains of the LLMs and task-specific small models, and (2) error accumulation arising from the predictions of both models. To address these challenges, we then introduce two key components in CoLLaTe: a model alignment module and a collaborative loss function. Through theoretical analysis and experimental validation, we demonstrate that these components effectively mitigate the identified challenges and achieve better performance than both LLM-based and task-specific models.

## 1 Introduction

Recently, methods based on Large Language Models (LLMs) have demonstrated the capablility of reading professional documents, effectively acquiring human knowledge to guide relevant tasks. However, they are often insensitive to the value fluctuations in time series data, and their natural language processing-based representations do not align well with the characteristics of time series data (Jin et al., 2024). In contrast, task-specific small scaled models, such as task-specific anomaly detection models (TSADM), typically lack the capabilities of reading professional documents and extracting expert knowledge. However, these models are specifically designed to learn patterns for anomaly detection and often exhibit superior performance when applied to well-matched anomaly detection datasets (Zhou et al., 2023). Despite their strengths, TSADMs also have notable limitations. First, for complex systems, such as cloud service system and flight systems, which require lots of expert knowledge to detect and diagnose anomalies accurately, the anomaly assumptions of unsupervised anomaly detection methods can be inconsistent with these scenarios (Zha et al., 2020). For instance, anomalies identified based on the voting mechanism of redundant aircraft channels (Rice & Mccorkle, 1979) may differ from the outlier assumptions utilized by unsupervised anomaly detection methods (Blázquez-García et al., 2021). Thus, researchers need to modify TSADMs to incorporate the background expertise to achieve optimal performance. For instance, anomaly detection methods tailored for cloud service monitoring (Ma et al., 2021; Chen et al., 2024b) or aircraft monitoring (e Silva & Murcca, 2023) have been modified to suit these specific contexts. Second, in many practical scenarios, such as aircraft monitoring (Nanduri & Sherry, 2016), it is not feasible to collect system monitoring data for all possible flight conditions. These conditions often represent distinct distributions and normal patterns in the monitoring data. This insufficient coverage of comprehensive data poses a significant challenge, greatly impeding the performance of TSADM.

Inspired by the human nervous system, which relies on the brain to store expert knowledge and extract general principles, while using the peripheral nervous system and spinal cord for specific tasks like the withdrawal reflex and knee-jerk reflex, we propose a framework called CoLLaTe. This framework

aims to facilitate effective **Co**llaboration between a **L**arge **La**nguage model (analogous to the brain) and a **T**ask-sp**e**cific model (analogous to the peripheral nervous system and spinal cord) to leverage their complementary strengths for the anomaly detection task. By enabling a strong synergy between an LLM and a TSADM, we can conveniently embed expert domain knowledge by adjusting the prompt of LLM, including necessary background knowledge and eliminating the need to design specialized models for different application scenarios. This approach also mitigates performance degradation caused by insufficient monitoring data across diverse operational conditions, because LLMs excel at utilizing professional documents to incorporate domain knowledge, which is often expressed in natural language. Such knowledge is highly condensed and can cover more general situations compared to concrete data samples. For instance, compared with providing different time series and describing whether they are trigonometric or not, giving the rule $a\sin(w_1 x + t_1) + b\cos(w_2 x + t_2)$ is more condensed and can represent more general situations. Thus, in the cases where training data samples are scarce, LLMs can extract general standards or patterns from professional documents to guide anomaly detection and mitigate the performance degradation on unseen distributions.

To this end, CoLLaTe integrates an LLM, enhanced by professional documents, with a TSADM to assess the severity of anomalies for each time slot. Subsequently, CoLLaTe employs a conditional network to synthesize the judgments from the LLM and the TSADM, using the data representation from the TSADM as a condition, to produce a unified anomaly score. As illustrated in Fig. 1(a), the collated anomaly score retains the true positive judgments from both the LLM and the TSADM while reconciling false positives. During this collaborative process, two main challenges arise:

- *Misalignment between the expression domains of the LLM and the TSADM*. The LLM and the TSADM interpret anomaly scores differently, meaning they may use the same score to represent different levels of anomaly severity. In the example shown in Fig. 1(b), most anomaly scores produced by the LLM are below 0.3, whereas the anomaly scores generated by the TSADM are predominantly centered around 0.4 after normalization. Consequently, an anomaly score of 0.4 might indicate a moderate anomaly for the LLM, but for the TSADM, it could signify a normal condition. Such inconsistencies in score interpretation can disrupt effective collaboration between the two models.

- *Prediction error accumulation*. Both the LLM and the TSADM are subject to prediction errors. During the collaboration process, these errors may not cancel out but instead accumulate. Through theoretical analysis and experimental validation, we show that when using classical loss functions, such as Mean Squared Error (MSE), the errors from the two models tend to either compound or settle at a compromise between the higher and lower values, rather than canceling each other out.

To enable effective synergy between the LLM and the TSADM while addressing the challenges mentioned above, we make the following key contributions:

- **The CoLLaTe framework**. We first formalize the collaboration process between the LLM and the TSADM, identifying the key challenges that arise during this process. To address these challenges, we propose the CoLLaTe framework that facilitates seamless collaboration between the LLM and the TSADM.

- **LLM and TSADM alignment**. We introduce a novel alignment module to harmonize the differing interpretations of anomaly scores between the LLM and the TSADM.

- **Theoretically sound collaborative loss function**. We conduct a theoretical analysis of the aforementioned error accumulation phenomenon and propose a novel provably collaborative loss function to mitigate it.

- We conduct extensive experiments to validate the effectiveness of CoLLaTe and each of its proposed modules.

## 2 METHOD

### 2.1 OVERVIEW

The architecture of CoLLaTe is depicted in Fig. 2. The anomaly scores, $\dot{s}$ and $S$, are generated by a TSADM and an LLM, respectively, where a set-up pitch prompt is utilized to embed expert knowledge

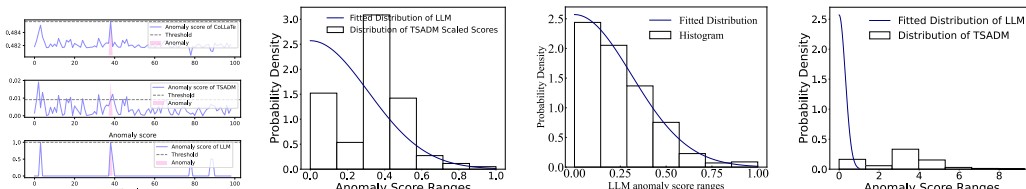

(a) The performance of CoLLaTe, LLM and TSADM

(b) Misalignment between anomaly scores of the LLM and TSADM

(c) Anomaly score distribution of large language model

(d) Misalignment between original anomaly scores of LLM and TSADM

Figure 1: (a) The anomaly scores of LLM, TSADM, and CoLLaTe. (b) The normalized anomaly score distribution of TSADM and the fitted curve of anomaly scores from the LLM on Mustang (Amvrosiadis et al., 2018). (c) shows the histogram of the anomaly score of LLM on the Mustang (Amvrosiadis et al., 2018) dataset and the fitted curve of the anomaly score. (d) The original anomaly score distribution of TSADM for anomaly detection and the fitted curve of anomaly scores from the LLM on Mustang.

from professional document to the LLM (Appendix. A.12). The TSADM is an anomaly attention (Xu et al., 2022) based anomaly detection model, whose architecture is illustrated in Appendix. A.10 in detail. These anomaly scores, $\dot{s}$ and $S$, typically represent different score interpretations, as the LLM and the TSADM can assign the same score to denote varying degrees of anomaly. To address this, we introduce an alignment module to align the different interpretations of the anomaly scores between the LLM and the TSADM. Following alignment, we employ a conditional network to synthesize the aligned scores from the TSADM and anomaly scores from the LLM, using the data representation $R$ as a condition. This process yields the collated anomaly score, $\hat{S}$, which is deemed as the final anomaly score to distinguish anomaly. Additionally, we propose a collaborative loss function that effectively leverages the complementary strengths of the LLM and the TSADM. Notably, we prove this loss function can mitigate error accumulation during the collaborative process between the LLM and the TSADM mathematically and experimentally.

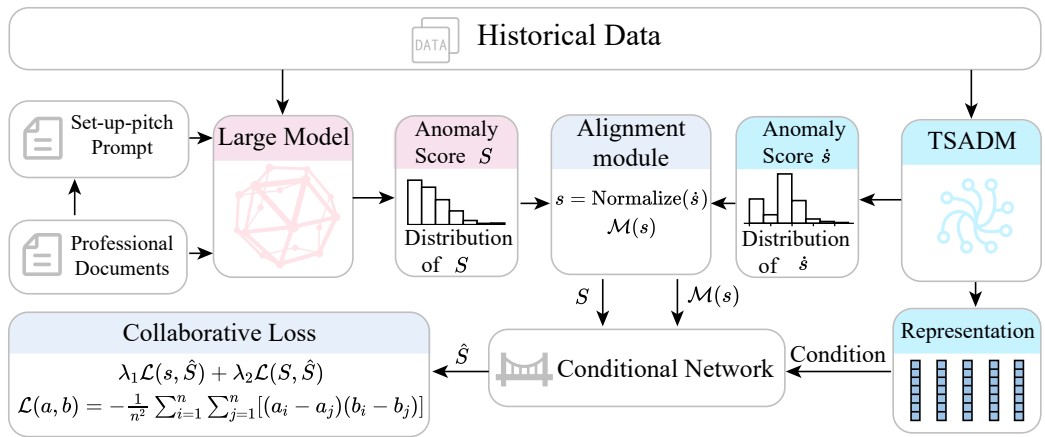

Figure 2: The model architecture of CoLLaTe

## 2.2 ALIGNMENT MODULE

TSADM and large language models (LLMs) often interpret anomaly scores differently. In other words, they may assign the same anomaly score to represent varying degrees of anomaly severity. This discrepancy arises due to two main factors: value range misalignment and distribution misalignment. The value range misalignment is shown in Fig. 1(d), where we shift the fitted curve of LLM anomaly scores to the histogram of TSADM anomaly scores. Most reconstruction-based TSADMs use reconstruction error as the anomaly score, resulting in a highly variable range of scores. In contrast, the anomaly scores produced by LLMs are typically constrained to the range $[0, 1]$. This problem can be easily solved by min-max normalization and we obtain the scaled anomaly scores of TSADM, $s$. This process is illustrated in detail in Appendix. A.13.

Table 1: The average F1 score of TSADM and LLM on datasets[1].

| | Flight1 | | Flight2 | | Mackey | | Mustang | |
|---|---|---|---|---|---|---|---|---|
| | Contextual | Point | Contextual | Point | Long | Short | Long | Short |
| TSADM | **0.694** | 0.093 | **0.561** | 0.061 | **0.848** | 0.162 | **0.931** | 0.192 |
| LLM | 0.630 | **0.174** | 0.409 | **0.231** | 0.733 | **0.199** | 0.556 | **0.399** |

Even after scaling the anomaly scores of the TSADM, differences in the distribution of anomaly scores can still lead to inconsistent interpretations. As shown in Fig. 1(c), we use a half-Gaussian distribution, which will be elaborated in the following, to fit the histogram of LLM anomaly scores. Then, we shift the fitted curve to the histogram of the anomaly scores from the TSADM in Fig. 1(b), where we observe that most LLM anomaly scores fall below 0.3, whereas the TSADM's anomaly scores are concentrated in the range of [0.3, 0.4]. As a result, an anomaly score of 0.4 might indicate moderate anomaly risks for the LLM, while the TSADM may interpret it as normal. In such cases, direct collaboration between the TSADM and the LLM without proper alignment could misrepresent their intended outputs. For example, if the LLM assigns a score of 0.4 to a time slot, the TSADM might incorrectly assume the LLM considers the time slot normal. To address this challenge, we propose an Alignment Module to align their interpretations and ensure effective collaboration.

Inspired by university grading policies, which align scores across professors by setting proportions for each grade level (e.g., controlling pass and excellence rates), we design the Alignment Module to align the distributions of anomaly scores from the LLM and the TSADM. This ensures that the same anomaly score represents a similar degree of anomaly for both models. To achieve this, we use a derivable function $f(S)$, which has a close format, to fit the LLM's anomaly score distribution (also guided by Reviewer ct8Q and Reviewer 2vmL).

Subsequently, we use a mapping function $\mathcal{M}(s)$ to map $s \in \{s_i\}_{i=0}^n$ from the TSADM to a new set, whose distribution should approach $f(S)$. To achieve this, we divide the value range of anomaly scores, $[0, 1]$, into N bins. We use $c_i$ to denote how many mapped scores $\mathcal{M}(s)$ fall into the $i^{th}$ bin. Then, we minimize cross entropy to decrease the distance between $f(S)$ and the distribution mapped set as Eq. 1 shows, where $\hat{\mu}, \hat{\sigma}$ are the mean and standard variance of distribution $f(S)$, $\mu_M = \frac{1}{n}\sum_{k=1}^n \mathcal{M}(s_k)$, and $\hat{\lambda}_1$ and $\hat{\lambda}_2$ are hyperparameters and bigger than zero. The first term in Eq. 8 is the cross-entropy, while the second and third terms are penalty terms that encourage the aligned distribution of the small model's anomaly scores to match the mean and variance of the large model's anomaly score distribution. The problem with Eq. 1 is that the counting process of $c_i$ is non-differentiable and could not be used in backforward propagation.

$$\min_{\mathcal{M}} . -\sum_{i=1}^N \frac{c_i}{\sum_{j=1}^N c_j} \log\left[\int_{(i-1)/N}^{i/N} f(S)dS\right] + \hat{\lambda}_1\left(\frac{1}{n}\sum_{i=1}^n \mathcal{M}(s_i) - \hat{\mu}\right)^2$$
$$+ \hat{\lambda}_2\left(\frac{1}{n-1}\sum_{i=1}^n (\mathcal{M}(s_i) - \mu_M)^2 - \hat{\sigma}^2\right)^2 \quad (1)$$

To solve the above problem, we prove that when $N$ approaches infinite, the objective function in Eq. 1 is equivalent to $-\frac{1}{n}\sum_{i=1}^n \log(f(\mathcal{M}(s_i))) + \hat{\lambda}_1(\frac{1}{n}\sum_{i=1}^n \mathcal{M}(s_i) - \hat{\mu})^2$, which is differentiable. The proof is given in Appendix. A.1. Consequently, we use Eq. 2 as our loss function.

$$\min_{\mathcal{M}} . \mathcal{L}_a = -\frac{1}{n}\sum_{i=1}^n \log(f(\mathcal{M}(s_i))) + \hat{\lambda}_1\left(\frac{1}{n}\sum_{i=1}^n \mathcal{M}(s_i) - \hat{\mu}\right)^2 + \hat{\lambda}_2\left(\frac{1}{n-1}\sum_{i=1}^n (\mathcal{M}(s_i) - \mu_M)^2 - \hat{\sigma}^2\right)^2$$
$$(2)$$

## 2.3 COLLABORATIVE LOSS

As mentioned before, we introduce a conditional network to synthesize the assessment of the TSADM and LLM. This network takes the aligned anomaly scores from the two models as input, along with the input data representation $R$ obtained from the TSADM as a condition. It outputs a collated anomaly score, as shown in Eq. 3, where $\mathcal{P}$ represents the trainable parameters of the conditional network. To optimize the collated anomaly scores in an unsupervised training process, we propose a collaborative loss function to leverage the complementary strength of LLM and TSADM.

---

[1]The datasets used in this table are introduced in Appendix. A.7

$$\hat{S} = \text{ConditionalNet}(\mathcal{M}(s), S, R; \mathcal{P}) \tag{3}$$

Many anomaly detection studies categorize anomalies into contextual anomalies and point anomalies (Schmidl et al., 2022; Tuli et al., 2022). We conducted experiments on two aircraft monitoring datasets and two public benchmarks, which are detailed in Appendix A.7. Since point anomalies in the two public benchmarks are few, we calculate the short deviation instead of point anomaly in these two datasets. The results, presented in Tab. 1, demonstrate the complementary performance of the TSADM and the LLM in detecting contextual and point anomalies. Specifically, the F1 score for contextual anomalies is higher for the TSADM compared to GPT-4, while GPT-4 achieves a better F1 score for point anomalies than the TSADM. According to this insight, we propose a collaborative loss function that adaptively optimizes the collated anomaly score. Specifically, we begin by dividing the time series $x \in R^{T \times D}$ into patches $P \in R^{t \times T/t \times D}$(Nie et al., 2023). For each time slot, we compute the average distance between the present time slot and all other time slots within the same patch, denoted as $D_{intra}$, and calculate the average distance between the patch of the current time slot and all other patches, denoted as $D_{inter}$, which are defined formally in Eq. 4. $d(\cdot, \cdot)$ is defined as the Euclidean distance and $k$ is the index of the patch to which the $i$-th time step belongs. (Also guided by Reviewer ct8Q)

$$D_{inter}(i) = \frac{1}{T/t - 1} \sum_{j \in [0, T/t] \setminus i} d(P[k][i], P[k][j]), D_{intra}(i) = \frac{1}{t - 1} \sum_{j \in [0, t] \setminus i} d(P[i], P[j]) \tag{4}$$

Since anomalous time slots differ significantly from normal ones, while normal time slots tend to share similarities, the metrics $D_{intra}$ and $D_{inter}$ behave differently for point and contextual anomalies. For point anomalies, $D_{intra}$ should be large because the point anomaly stands out distinctly from the surrounding normal time slots. Conversely, $D_{inter}$ for point anomalies should be smaller, as the normal time slots within the current patch resemble those in other patches, reducing the average difference. For contextual anomalies, which typically involve a sustained deviation, the anomalies within an anomalous event are similar to each other but differ significantly from normal time slots. As a result, $D_{intra}$ should be small, while $D_{inter}$ should be large. Considering LLM can handle point anomaly better and TSADM can handle contextual anomaly better, we design the collaborative loss function in Eq. 5, where $\mathcal{L}(a, b)$ represents a loss function that measures the distance between $a$ and $b$.

$$\mathcal{L}_\beta = \frac{D_{intra}}{D_{intra} + D_{inter}} \mathcal{L}(s, \hat{S}) + \frac{D_{inter}}{D_{intra} + D_{inter}} \mathcal{L}(S, \hat{S}) \tag{5}$$

If we choose some popular loss functions as $\mathcal{L}(a, b)$, such as $\mathcal{L}(a, b) = (a - b)^2$, $\mathcal{L}_\beta$ will accumulate the prediction error of TSADM and LLM in the process of gradient descent, as shown in Theorem 1, which is proven in Appendix. A.2.

**Assumption 1.** The anomaly score prediction error of the TSADM is $\epsilon_s$, where $\epsilon_s$ obeys an unknown distribution $\mathcal{D}_s(\mu_s, \sigma_s)$, $\mu_s \neq 0$.

**Assumption 2.** The anomaly score prediction error of the LLM is $\epsilon_S$, which obeys an unknown distribution $\mathcal{D}_S(\mu_S, \sigma_S)$, $\mu_S \neq 0$.

$$\mathcal{L}_\beta^* = -\sum_{i=1}^{n} \sum_{j=1}^{n} [(y_i - y_j)(\hat{S}_i - \hat{S}_j)] \tag{6}$$

We examined the validity of these Assumptions in the context of our experiment through empirical discussion, which is shown in Appendix. A.16.

**Theorem 1.** Given Assumption 1 - Assumption 2, when $\mathcal{L}(a, b) = (a - b)^2$, the difference between the optimal solution $\hat{S}^*$ of loss function $\mathcal{L}_\beta$ and ground truth $y$ is $\mathbb{E}[(\hat{S}^* - y)^2] \geq (\lambda_1 \mu_s + \lambda_2 \mu_S)^2$, where $\lambda_1 = \frac{D_{intra}}{D_{intra} + D_{inter}}$ and $\lambda_2 = \frac{D_{inter}}{D_{intra} + D_{inter}}$.

From Theorem 1, we observe that when MSE is used as $\mathcal{L}$, the expected prediction error of the conditional network exceeds the weighted sum of the expected prediction errors of the TSADM and LLM. Given that the weights $\lambda_1$ and $\lambda_2$ are both greater than 0 and sum to 1, it indicates that the collaborative error, when using MSE as $L_{(a,b)}$, consistently exceeds the smaller of the errors produced by the LLM and TSADM. This phenomenon reflects error accumulation and is further validated experimentally, as discussed in detail in Section 3.4.

To solve this problem we propose the collaborative loss function as shown in Eq. 7, where $\hat{S}_i$, $s_i$ and $S_i$ are anomaly scores of $i^th$ sample output by the conditional network, TSADM and LLM respectively. In Lemma 1, we prove that using the collaborative loss function is approximate to using the loss function in Eq. 6. This equivalence shows that Eq. 7 prevents error accumulation between the TSADM and LLM, as Eq.6 directly minimizes the difference between the collated anomaly scores of two samples and the ground truth. Additionally, we experimentally confirm that Eq. 7 effectively avoids error accumulation, as discussed in Section. 3.4.

$$\mathcal{L}_\beta = -\frac{1}{n^2} \sum_{i=1}^{n} \sum_{j=1}^{n} \lambda_1[(s_i - s_j)(\hat{S}_i - \hat{S}_j)] \tag{7}$$
$$+ \lambda_2[(S_i - S_j)(\hat{S}_i - \hat{S}_j)]$$

**Lemma 1.** When Assumption 1 - Assumption 3 hold, as the iteration step $T$ approaches infinity, minimizing Eq. 7 by stochastic gradient descent can be approximate to minimizing Eq. 6 by stochastic gradient descent with convergence rate $\mathbf{O}(T^{-\frac{1}{4}})$.

The proof of Lemma 1 is given in Appendix. A.3.

Moreover, in Theorem 2, we prove that the optimal solution of Eq. 6 satisfies two key properties. Property 1 denotes that if the anomaly degree of time slot $r$ is more serious than the one of time slot $k$, the optimal collated anomaly score of time slot $r$ is bigger than that of time slot $k$. Property 2 denotes that if the difference in anomaly severity between time slots $r$ and $k$ is greater than the difference between time slots $r'$ and $k'$, the difference in optimal collated anomaly scores will accurately reflect this gap.

**Theorem 2.** When Assumption 1 - Assumption 3 hold, the optimal solution $\hat{S}^*$ of the loss function in Eq. 6 satisfies the following properties.
*Property 1.* $\forall r, \forall k$, if the ground truth $y_r > y_k$, then $\hat{S}_r^* > \hat{S}_k^*$, where $y_r$ and $y_k$ denotes the ground truth of the anomaly scores of $r^{th}$ and $k^{th}$ samples respectively, $\hat{S}_r^*$ and $\hat{S}_k^*$ are the optimal solution for Eq. 6 of $r^{th}$ and $k^{th}$ samples respectively.
*Property 2.* $\forall r, \forall k, \forall r', \forall k'$, if the ground truth $y_r - y_k > y_{r'} - y_{k'}$, then $\hat{S}_r^* - \hat{S}_k^* > \hat{S}_{r'}^* - \hat{S}_{k'}^*$.

The proof of Theorem 2 is given in Appendix. A.4.

## 3 EXPERIMENT

We make extensive experiments on four datasets and make the following contributions:

- CoLLaTe can achieve the best performance compared with SOTA LLM-based and TSADMs.
- CoLLaTe can maintain good performance on unseen distributions.
- We locate the hyperparameters important to CoLLaTe performance and analyze their effect.
- All the modules in CoLLaTe are effective and contribute to its performance.

### 3.1 EXPERIMENT SETUP

**Baselines**. We compare CoLLaTe with state-of-the-art (SOTA) anomaly detection methods, including DCdetector (Yang et al., 2023), AnomalyTransformer (Xu et al., 2022), TranAD (Tuli et al., 2022), OmniAnomaly (Su et al., 2019), and MSCRED (Zhang et al., 2019), Time Series Foundation Model, Timer (Liu et al., 2024c), Moment (Goswami et al., 2024), OneFitsAll (Zhou et al., 2023). Additionally, we evaluate against SOTA LLM-based anomaly detection methods, including GPT-4, Qwen3, LLMAD (Liu et al., 2024b), and sigLLM (Alnegheimish et al., 2024).

**Datasets.** To verify CoLLaTe can effectively embed the expertise knowledge, we choose three datasets for a complex system, which has domain-specific rules for anomaly detection and highly rely on expert knowledge. One is cluster-wide slowdown detection in cloud environment (Mustang) (Amvrosiadis et al., 2018) and two are aircraft monitoring datasets collected from an aircraft manufacturing company that is one of the world's top 150 enterprises. Additionally, we included a subtle mathematical-assumption-based anomaly detection dataset Mackey (Thill & Konen, 2020), in which anomalies

Table 2: Performance comparison of CoLLaTe and all baselines on Mustang, Mackey, Flight1, and Flight2 datasets. The best results are in **bold**, and the second-best are underlined.

| | Mustang | | | Mackey | | | Flight1 | | | Flight2 | | |
|---|---|---|---|---|---|---|---|---|---|---|---|---|
| | Prec | Rec | F1 | Prec | Rec | F1 | Prec | Rec | F1 | Prec | Rec | F1 |
| GPT4 | 0.450 | 0.797 | 0.448 | 0.813 | 0.970 | 0.862 | 0.754 | 0.673 | 0.620 | 0.322 | **1.000** | 0.422 |
| LLMAD | 0.068 | 0.800 | 0.122 | 0.077 | 0.696 | 0.132 | 0.138 | 0.982 | 0.226 | 0.171 | 0.873 | 0.274 |
| sigLLM | - | - | - | 0.063 | 0.589 | 0.107 | - | - | - | - | - | - |
| MSCRED | 0.871 | 0.960 | 0.896 | 0.829 | 0.960 | 0.882 | 0.690 | 0.947 | 0.768 | 0.502 | 0.987 | 0.595 |
| OmniAnomaly | 0.812 | 0.968 | 0.878 | 0.761 | 0.995 | 0.822 | 0.619 | **1.000** | 0.721 | 0.579 | **1.000** | 0.637 |
| TranAD | 0.865 | 0.918 | 0.867 | 0.301 | 0.500 | 0.367 | 0.340 | **1.000** | 0.497 | 0.225 | **1.000** | 0.367 |
| AnomalyTransformer | **1.000** | 0.891 | 0.935 | 0.934 | 0.807 | 0.851 | 0.750 | 0.649 | 0.685 | 0.549 | 0.469 | 0.484 |
| DCdetector | 0.968 | 0.718 | 0.799 | 0.200 | 0.200 | 0.200 | 0.759 | 0.615 | 0.664 | 0.749 | 0.742 | 0.746 |
| Qwen3-235B | 0.483 | 0.362 | 0.414 | 0.442 | 0.986 | 0.610 | 0.833 | 0.500 | 0.625 | 0.457 | 0.889 | 0.604 |
| OneFitsAll | 0.366 | 0.318 | 0.340 | 0.355 | 0.524 | 0.423 | 0.133 | **1.000** | 0.235 | 0.261 | 0.546 | 0.353 |
| Timer | - | - | - | 0.242 | 0.541 | 0.334 | - | - | - | - | - | - |
| Moment | 0.054 | **1.000** | 0.102 | 0.086 | 0.919 | 0.157 | 0.100 | **1.000** | 0.182 | 0.111 | **1.000** | 0.200 |
| w/o alignment module | 0.914 | 1.000 | 0.953 | 0.927 | **0.996** | 0.959 | 0.606 | **1.000** | 0.729 | 0.682 | 0.762 | 0.627 |
| w/o collaborative loss | 0.615 | 0.933 | 0.738 | 0.490 | 0.996 | 0.578 | 0.675 | 0.830 | 0.667 | 0.416 | 0.991 | 0.529 |
| w/o $\lambda_1, \lambda_2$ | 0.968 | 0.933 | 0.943 | 0.522 | 0.970 | 0.594 | 0.631 | 0.894 | 0.648 | 0.436 | 0.755 | 0.530 |
| w/o LLM | 0.950 | 0.943 | 0.938 | 0.925 | **0.996** | 0.957 | 0.566 | **1.000** | 0.694 | 0.775 | 0.819 | 0.702 |
| CoLLaTe-HalfGaussian | **1.000** | 0.943 | 0.967 | **0.970** | 0.996 | **0.982** | 0.790 | 0.978 | 0.866 | 0.897 | 0.891 | 0.883 |
| CoLLaTe-GMM | 0.980 | **1.000** | **0.989** | 0.962 | **0.996** | 0.978 | **0.805** | 0.978 | **0.872** | **0.835** | 0.978 | **0.889** |

Table 3: The average performance of CoLLaTe and baselines on the unseen datasets. The best results are in **bold**, and the second-best are underlined.

| | Mustang | | | Mackey | | | Flight1 | | | Flight2 | | |
|---|---|---|---|---|---|---|---|---|---|---|---|---|
| | Prec | Rec | F1 | Prec | Rec | F1 | Prec | Rec | F1 | Prec | Rec | F1 |
| MSCRED | 0.719 | **0.984** | 0.804 | 0.671 | 0.963 | 0.765 | 0.734 | 0.956 | 0.765 | 0.426 | 0.983 | 0.521 |
| OmniAnomaly | 0.483 | 0.956 | 0.605 | 0.823 | 0.995 | 0.881 | 0.329 | 0.964 | 0.455 | 0.223 | 0.948 | 0.285 |
| TranAD | 0.308 | 0.501 | 0.354 | 0.321 | 0.600 | 0.395 | 0.188 | 0.600 | 0.282 | 0.090 | 0.400 | 0.147 |
| AnomalyTransformer | 0.200 | 0.200 | 0.200 | **0.976** | 0.667 | 0.765 | 0.391 | 0.251 | 0.305 | 0.592 | 0.458 | 0.491 |
| DCdetector | 0.250 | 0.250 | 0.250 | 0.325 | 0.399 | 0.353 | 0.408 | 0.295 | 0.325 | 0.750 | 0.747 | 0.748 |
| CoLLaTe-HalfGaussian | 0.950 | 0.943 | 0.938 | 0.916 | **0.996** | 0.948 | **0.796** | **1.000** | **0.877** | **0.897** | 0.891 | **0.883** |
| CoLLaTe-GMM | **0.953** | 0.944 | **0.949** | 0.928 | **0.996** | 0.958 | 0.742 | **1.000** | 0.840 | 0.717 | **1.000** | 0.820 |

can not be recognized visually, but need complex mathematical derivation and verification. For more details, please refer to Appendix A.7.

**Hyperparameters.** We show the hyperparameter values in Appendix A.8.

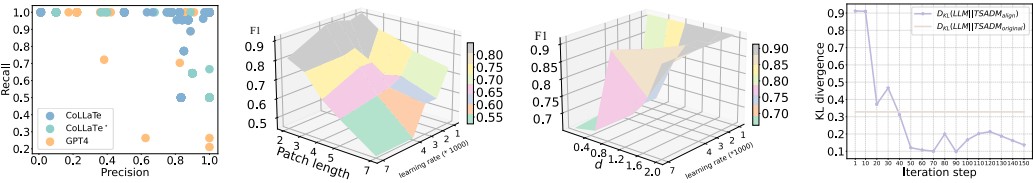

(a) Performance of CoL-LaTe, TSADM and LLM  (b) The impact of patch length and learning rate  (c) The impact of $d$ and learning rate  (d) KL distance between the anomaly scores of the two models

Figure 3: (a) We use coordinates (Precision, Recall) to denote performance on a subset of flight dataset. We draw a scatter plot to show the distribution of LLM (GPT4), TSADM (CoLLaTe$^\star$), and CoLLaTe; (b) The figure shows changes in F1 score as patch length in $D_{inter}$, $D_{intra}$ and learning rate vary, where the learning rate ticks are the original value multiplied by 1000 times; (c) The figure shows changes in F1 score as $d$ and learning rate vary, where the learning rate ticks are the original value multiplied by 1000 times; (d) The figure shows that KL-divergence between the histogram of LLM anomaly score and aligned TSADM anomaly score and compares it with the KL-divergence between LLM anomaly score and original TSADM anomaly score;

**Evaluation Metrics.** We use three of the most popular evaluation metrics as many works did (Chen et al., 2022b; 2021; Li et al., 2023): the recall, precision and F1 score.

## 3.2 PREDICTION ACCURACY

Precision, recall, and F1 score are calculated for each subset, and the average results are presented in Tab. 2, where the best performance is highlighted in bolded, and if CoLLaTe achieves the best performance we underline the best performance among baselines. In the table, "prec" and "rec" denote precision and recall, while "OmniAnom" and "AnomalyTr" represent "OmniAnomaly" and "AnomalyTransformer", respectively. As sigLLM is specifically designed for univariate time series, it cannot be applied to the Mustang, Flight 1 and Flight 2 datasets, and therefore, its performance on these datasets is not included. In Tab. 2, We test the performance of CoLLaTe under different distribution assumptions: Gaussian Mixture Model and Half-Gaussian. The goodness-of-fits test and CoLLaTe performance under other assumptions can be found in Appendix. A.19. As shown in Tab. 2, CoLLaTe achieves the highest F1 scores across all four datasets. In this table, GPT-4 represents the LLM method employing the set-up-pitch prompt, which is the LLM component of CoLLaTe. By comparing the performance of GPT-4 with other LLM-based anomaly detection methods, such as sigLLM and LLMAD, it becomes evident that the set-up-pitch prompt significantly enhances the anomaly detection accuracy of the LLM. In Fig. 3(a), we plot the precision and recall corresponding to the best F1 scores as coordinates to illustrate the performance distribution of CoLLaTe, GPT-4, and CoLLaTe$^\star$. The figure reveals that both the LLM and the TSADM (CoLLaTe$^\star$) tend to suffer from either low precision or low recall. In contrast, CoLLaTe demonstrates strong performance on both metrics. This observation confirms that CoLLaTe enables effective collaboration between the TSADM and the LLM, resulting in superior overall performance.

Since the datasets consist of different subsets and different subsets represent monitoring data for different cloud servers and aircraft components, they exhibit different distributions. To verify that LLM component in CoLLaTe can promote its performance on unseen distribution by leveraging general expertise knowledge, in Tab. 3, for each subset, we train the models on other subset and test the model on it. In this situation, CoLLaTe can maintain its strength over other methods, even for some baselines like TranAD equipping with meta-learning to improve its generalization. Thus, LLM component in CoLLaTe can effectively leverage the general expertise knowledge. Since we directly use pre-trained models of LLM-based methods and TSFM, their performances are same in Tab. 2 and Tab. 3 and are only shown once.

## 3.3 HYPERPARAMETER SENSITIVITY

We investigate the impact of several factors on CoLLaTe's performance, including $d$ in Eq.37, patch size for $D_{inter}$ and $D_{intra}$, learning rate, and $\hat{\lambda}_1$ and $\hat{\lambda}_2$ in Eq.2. Notably, when $\hat{\lambda}_1$ and $\hat{\lambda}_2$ vary within $[0.01, 0.9]$, the change in CoLLaTe's F1 score is within 0.01. Thus, our analysis primarily focuses on the effects of $d$, patch size, and learning rate. In Fig.3(b), we examine CoLLaTe's F1 score as the patch size of $D_{inter}$ and $D_{intra}$ varies from 2 to 7, and the learning rate changes from 0.001 to 0.007. As shown in Fig.3(b), the F1 score improves as both the patch size and learning rate decrease. CoLLaTe performs better with smaller patch sizes because larger patch sizes that exceed the length of contextual anomalies lead to smaller $D_{inter}$. In such cases, since both contextual anomalies and point anomalies have small $D_{inter}$ and large $D_{intra}$, distinguishing between these anomaly types becomes difficult, thereby degrading CoLLaTe's performance. In Fig.3(c), we evaluate CoLLaTe's F1 score as $d$ ranges from 0.01 to 2 and the learning rate varies from 0.001 to 0.007.

## 3.4 EFFECTIVENESS OF EACH MODULE

**Ablation experiment**. In CoLLaTe$^\dagger$, we remove the Alignment module. In CoLLaTe$^\ddagger$, we change $\mathcal{L}(a, b)$ to MSE. In CoLLaTe$^*$, we change $\frac{D_{intra}}{D_{inter}+D_{intra}}$ and $\frac{D_{inter}}{D_{inter}+D_{intra}}$ to 1. We compare their performance with CoLLaTe in Tab. 2, where CoLLaTe outperforms all of them. This proves that each module in CoLLaTe contributes to its performance.

**Effectiveness of the collaborative loss**. CoLLaTe$^\star$ is the TSADM utilized within CoLLaTe. By comparing the performance of CoLLaTe, GPT-4, and CoLLaTe$^\star$, it is evident that the collaboration between the TSADM CoLLaTe$^\star$ and the LLM-based model GPT-4 allows CoLLaTe to outperform both.

This demonstrates that CoLLaTe effectively mitigates error accumulation during the collaboration between GPT-4 and CoLLaTe$^\star$ and coincides with the theoretical analysis in Theorem 1.

**Effective of the design of** $\mathcal{L}(a, b)$. Moreover, as mentioned earlier, CoLLaTe$^\ddagger$ refers to the method that replaces $\mathcal{L}(a, b)$ in the collaborative loss function with MSE. On the Mackey dataset, the performance of CoLLaTe$^\ddagger$ is worse than both GPT-4 and CoLLaTe$^\star$. On other datasets, its performance is consistently worse than one of GPT-4 or CoLLaTe$^\star$ while being only slightly better than the other. This observation supports the analysis in Section 2.3, which indicates that using MSE as the formulation of $\mathcal{L}(a, b)$ leads to error accumulation between the LLM and the TSADM during the collaboration process.

**Effectiveness of the adaptive hyperparameters**. In CoLLaTe$^*$, the adaptive hyperparameters $\frac{D_{intra}}{D_{inter}+D_{intra}}$ and $\frac{D_{inter}}{D_{inter}+D_{intra}}$ are set to 1. When comparing CoLLaTe$^*$ to GPT-4 and the TSADM CoLLaTe$^\star$, we observe that on most datasets (Mackey, Flight 1, Flight 2), the performance of CoLLaTe$^*$ falls between that of the LLM (GPT-4) and the TSADM (CoLLaTe$^\star$). This is because, without effectively leveraging the complementary strengths of the LLM and the TSADM, the collaborative performance of CoLLaTe$^*$ becomes a compromise, reflecting a balance between the better-performing and lower-performing models. This observation highlights the importance and effectiveness of the adaptive hyperparameters.

**Effectiveness of the alignment module.** The ablation experiment of CoLLaTe$^\dagger$ demonstrates that the alignment module contributes significantly to the performance of CoLLaTe. To verify that the alignment module effectively reduces the distribution distance between the aligned anomaly scores of the TSADM and the anomaly scores of the LLM, we compute the KL-divergence distance between the aligned scores of the TSADM and the anomaly score histogram of LLM as the iteration step grows. We present the results in Fig. 3(d). The figure also compares the KL-divergence distance between aligned TSADM anomaly scores and that of LLM with the KL-divergence distance between original TSADM anomaly scores and that of LLM. From Fig. 3(d), we observe that the distribution distance between the aligned TSADM anomaly scores and the LLM anomaly scores gradually decreases as the iteration steps increase, and falls below the distribution distance observed between the original anomaly scores of the two models, which verifies that the alignment module effectively aligns the distribution of anomaly scores between the TSADM and the LLM.

## 4 RELATED WORK

TSADM can be categorized into prediction-based methods (Malhotra et al., 2015; Hundman et al., 2018; Zong et al., 2018; Chen et al., 2022a), reconstruction-based methods (Chen et al., 2022b; You et al., 2022; Jiang et al., 2022; Shen et al., 2021; Tian et al., 2019), and Time Series Foundation model (Yeh et al., 2023; Goswami et al., 2024; Zhou et al., 2023; Rasul et al., 2023). Prediction-based methods are vulnerable to historical inference errors (Wang et al., 2024). In scenarios where it is impossible to observe or collect all normal patterns (e.g., airplane monitoring), reconstruction-based methods are ineffective. Time Series Foundation models have better generalization performance, while they suffer from overgeneralization problem.

LLM-based methods (Liu et al., 2024b; Russell-Gilbert et al., 2024; Liu et al., 2024a) introduce expert knowledge effectively through techniques like retrieval-augmented generation (RAG) (Lewis et al., 2020) and chain of thought (COT) (Wei et al., 2022) without requiring model modifications. However, LLMs are insensitive to value fluctuations and normal pattern extraction.

## 5 CONCLUSION

Given the complementary strengths of LLMs and TSADM, where LLMs can conveniently incorporate expert knowledge, while TSADM are more sensitive to value fluctuations and normal pattern extraction, we propose a novel approach inspired by the human nervous system to facilitate their collaboration. We begin by identifying the key challenges in facilitating this collaboration and then introduce a collaborative framework, called CoLLaTe, to address them. There are two distinctive characteristics of CoLLaTe: an alignment module aligning the expression domain of LLM and TSADM, and a collaborative loss function reducing the error accumulation. We have made solid mathematical proof and extensive experiments to verify the effectiveness of the proposed methods.

## 6 REPRODUCIBILITY STATEMENT

We submit our code and data in supplementary files. Besides, we discuss the hyperparameter settings, detailed model architecture and prompt designing in Appendix. A.8, Appendix. A.9, Appendix. A.10, Appendix. A.12.

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

# A APPENDIX

## A.1 PROOF OF TRANSFORMATION

$$\min. -\sum_{i=1}^{N} \frac{c_i}{\sum_{j=1}^{N} c_j} \log[\int_{(i-1)b/N}^{ib/N} f(S)dS] + \hat{\lambda}_1(\frac{1}{n}\sum_{i=1}^{n}\mathcal{M}(s_i) - \hat{\mu})^2 \tag{8}$$
$$+\hat{\lambda}_2(\frac{1}{n-1}\sum_{i=1}^{n}(\mathcal{M}(s_i) - \mu_M)^2 - \hat{\sigma}^2)^2$$

$$\min_{\mathcal{M}}.\mathcal{L}_a = -\frac{1}{n}\sum_{i=1}^{n}[\log(f(\mathcal{M}(s_i))) - \log N + \hat{\lambda}_1(\frac{1}{n}\sum_{i=1}^{n}\mathcal{M}(s_i) - \hat{\mu})^2 + \hat{\lambda}_2(\frac{1}{n-1}\sum_{i=1}^{n}(\mathcal{M}(s_i) - \mu_M)^2 - \hat{\sigma}^2)^2] \tag{9}$$

Because the Half Gaussian function $f(S)$ is derivable in $(0, 1]$, has a right derivation on $S = 0$ and $f(S) \geq \frac{2}{\sigma\sqrt{2\pi}\sqrt[2]{e}} > 0$, according to Proposition 1, Eq. 8 can approximate Eq. 9 when $N$ approach infinite with error upper bound of $o(1)$. For any $\epsilon > 0$, we can find an $N_0$, if $N > N_0$, the error between the smooth surrogate loss function and the original non-differential one is below $\epsilon$. Thus, when optimizing $\mathcal{M}$, we can set $N$ as a big enough constant, which allows the error below the acceptable error limit. Then, the $-\log N$ (the $\log\frac{1}{N}$ item) is a constant and will not impact the minimization process and the optimal solution of the objective function. Thus, we omit it and obtain the smooth objective function in Eq. 2.

*Convergence Guarantee.* The summation in our objective contains $n$ terms, where $n$ denotes the number of data samples, rather than the number of bins $N$. That is because for most of bins, $c_i = 0$ and can be omitted in our simplification process above. In the summation, each term is a finite scalar, so the objective is simply a finite sum of finite quantities. Therefore, the optimization objective is well-defined and guaranteed to be convergent.

**Proposition 1.** Suppose the following conditions hold:

- $f(S)$ is derivable in $(0, 1]$;
- $f(S)$ has a right derivation on $S = 0$;
- $f(S)$ has a lower bound $l_b$ for $S \in [0, 1]$, and $l_b > 0$.

Then, Eq. 10 approximates $\frac{1}{n}\sum_{i=1}^{n}\log(f(\mathcal{M}(s_i)))$ with error upper bound of $o(1)$.

$$-\sum_{i=1}^{N}\frac{c_i}{\sum_{j=1}^{N}c_j}\log[\int_{(i-1)b/N}^{ib/N}f(S)dS] \tag{10}$$

*Proof of Proposition 1.* When the $N$ approaches infinite, Eq. 10 is equal to Eq. 11.

$$\lim_{N\to\infty} -\sum_{i=1}^{N}\frac{c_i}{\sum_{j=1}^{N}c_j}\log(\frac{f(\frac{ib}{N})}{N} + o(\frac{1}{N})) \tag{11}$$

For each $c_i$, there will be two situations. One is that there is no $\mathcal{M}(s_j) = \frac{ib}{N}, j \in [1, n]$, then $c_i = 0$ and the item including $c_i$ can be omitted. Another one is that there are some $\mathcal{M}(s_j) = \frac{ib}{N}, j \in [1, n]$, then $c_i$ is equal to the number of times that $\frac{ib}{N}$ appears in $\{\mathcal{M}(s_j)|s_j \in \{s_j\}_{j=1}^{n}\}$. In this situation, $\frac{c_i}{\sum_{j=1}^{N}c_j}\log(\frac{f(\frac{ib}{N})}{N} + o(\frac{1}{N}))$ can be transformed to $\frac{c_i}{n}\log(\frac{f(\mathcal{M}(s_j))}{N} + o(\frac{1}{N}))$, since $\sum_{j=1}^{N}c_j = n$.

We cluster the mapping scores with the same value as a group, from which we can obtain $M$ groups with values of $\{\mathcal{M}_{\hat{i}}\}_{\hat{i}=1}^{M}$, where the size of $\hat{i}^{th}$ group is $c_{\hat{i}}$ (i.e. $\hat{i}b/N = \mathcal{M}_{\hat{i}}$). The $\{c_{\hat{i}}\}_{\hat{i}=1}^{M}$ consist of the non-zero part of $\{c_i\}_{i=1}^{N}$. Then, Eq. 11 can be transformed to Eq. 12.

$$\lim_{N\to\infty} -\sum_{\hat{i}=1}^{M}\frac{c_{\hat{i}}}{n}\log(\frac{f(\mathcal{M}_{\hat{i}})}{N} + o(\frac{1}{N})) \tag{12}$$

Eq. 12 is exactly equal to traversing all the $\mathcal{M}(s_i), s_i \in \{s_i\}_{i=1}^n$, sum them up as $\frac{1}{n}\sum_{i=1}^n \log(\frac{f(\mathcal{M}(s_i))}{N} + o(\frac{1}{N}))$, and combine the items with same value of mapping scores. Thus, Eq. 12 is euqal to Eq. 13.

$$\lim_{N\to\infty} -\sum_{i=1}^n \frac{1}{n}\log(\frac{f(\mathcal{M}(s_i))}{N} + o(\frac{1}{N})) \tag{13}$$

The error between Eq. 13 and Eq. 10 is shown in Eq. 14. Because $\mathcal{M}(s_i) \in [0,1]$, $f(\mathcal{M}(s_i)) \geq l_b > 0$. Thus, $o(\frac{1}{N}) \times \frac{N}{f(\mathcal{M}(s_i))} = o(1)$. Then, we can derive Eq. 14 to Eq. 15. Since $N$ is irrelevant to $-\frac{1}{n}\sum_{i=1}^n[\log(1 + o(1))]$, we can derive Eq. 15 to Eq. 16. Since $\log(1+x) \leq 2x$ when $x < \frac{1}{2}$, we can derive Eq. 16 to Eq. 17.

$$\left| \lim_{N\to\infty} -\frac{1}{n}\sum_{i=1}^n [\log(\frac{f(\mathcal{M}(s_i))}{N} + o(\frac{1}{N})) - \log(\frac{f(\mathcal{M}(s_i))}{N})] \right| \tag{14}$$

$$= \left| \lim_{N\to\infty} -\frac{1}{n}\sum_{i=1}^n [\log(1 + o(1))] \right| \tag{15}$$

$$= \left| -\frac{1}{n}\sum_{i=1}^n [\log(1 + o(1))] \right| \tag{16}$$

$$\leq 2o(1) \tag{17}$$

### A.2 PROOF OF THEOREM 1

When $\mathcal{L}(a,b) = (a-b)^2$, we can transform Eq. 5 to Eq. 18, where $\lambda_1 = \frac{D_{intra}}{D_{intra}+D_{inter}}$ and $\lambda_2 = \frac{D_{inter}}{D_{intra}+D_{inter}}$.

$$\lambda_1(y + \epsilon_s - \hat{S}(\mathcal{P}, \epsilon_s, \epsilon_S))^2 + \lambda_2(y + \epsilon_S - \hat{S}(\mathcal{P}, \epsilon_s, \epsilon_S))^2 \tag{18}$$

According to KKT condition (Boyd & Vandenberghe, 2004), the optimal solution of $\mathcal{L}_\beta$ should satisfy Eq. 19.

$$2\lambda_1(y + \epsilon_s - \hat{S}^*(\mathcal{P}, \epsilon_s, \epsilon_S))\frac{\partial \hat{S}^*(\mathcal{P}, \epsilon_s, \epsilon_S)}{\partial \mathcal{P}} + 2\lambda_2(y + \epsilon_S - \hat{S}^*(\mathcal{P}, \epsilon_s, \epsilon_S))\frac{\partial \hat{S}^*(\mathcal{P}, \epsilon_s, \epsilon_S)}{\partial \mathcal{P}} = 0 \tag{19}$$

As shown in Appendix A.9, the gradient of architecture of conditional network can not be equal to 0. Thus, $2\lambda_1(y + \epsilon_s - \hat{S}^*(\mathcal{P}, \epsilon_s, \epsilon_S)) + 2\lambda_2(y + \epsilon_S - \hat{S}^*(\mathcal{P}, \epsilon_s, \epsilon_S)) = 0$. Then, $\hat{S}^*(\mathcal{P}, \epsilon_s, \epsilon_S) = y + \lambda_1\epsilon_s + \lambda_2\epsilon_S$. Thus, $\mathbb{E}[(\hat{S}^*(\mathcal{P}, \epsilon_s, \epsilon_S) - y)^2] = \mathbb{E}[(\lambda_1\epsilon_s + \lambda_2\epsilon_S)^2]$. Since $Var(X) = \mathbb{E}[(X - \mathbb{E}(X))^2] = \mathbb{E}[X^2] - (\mathbb{E}[X])^2 \geq 0$, we have $\mathbb{E}[X^2] \geq (\mathbb{E}[X])^2$. Thus, $\mathbb{E}[(\lambda_1\epsilon_s + \lambda_2\epsilon_S)^2] \geq |\mathbb{E}[\lambda_1\epsilon_s + \lambda_2\epsilon_S]|^2$. Since $\epsilon_1 \sim \mathcal{D}_s(\mu_s, \sigma_s), \epsilon_2 \sim \mathcal{D}_S(\mu_S, \sigma_S)$, we have $|\mathbb{E}[\lambda_1\epsilon_s + \lambda_2\epsilon_S]|^2 = (\lambda_1\mu_s + \lambda_2\mu_S)^2$. Thus, $\mathbb{E}[(\hat{S}^*(\mathcal{P}, \epsilon_s, \epsilon_S) - y)^2] \geq (\lambda_1\mu_s + \lambda_2\mu_S)^2$

### A.3 PROOF OF LEMMA 1

When using stochastic gradient descent algorithm, the gradient of Eq. 5 and Eq. 6 is shown in Eq. 20 and Eq. 21 respectively, where $\mathcal{B}$ is the set of sample in batch and $|\mathcal{B}|$ is the size of set $\{(r,t)|r \in \mathcal{B}, t \in \mathcal{B}\}$. Let $\mathbf{g}_{i,j}^*$ denote the item in Eq. 21, whose index is $(i,j)$. Let $\mathbf{g}_{i,j}$ denote the item in Eq. 20, whose index is $(i,j)$. $\mathbf{g}_{i,j}$ can be transformed to $\mathbf{g}_{i,j}^* + [\lambda_1(\epsilon_{s,i} - \epsilon_{s,j}) + \lambda_2(\epsilon_{S,i} - \epsilon_{S,j})](\hat{S}_i - \hat{S}_j)\frac{\partial(\hat{S}_i - \hat{S}_j)}{\partial \mathcal{P}}$. Thus, the gradient of Eq. 5 can be deemed as the gradient of Eq. 6 add some random noise scaled by $(\hat{S}_i - \hat{S}_j)\frac{\partial(\hat{S}_i - \hat{S}_j)}{\partial \mathcal{P}}$ for each sample.

$$\mathbf{g} = -\frac{1}{|\mathcal{B}|}\sum_{(i,j)\in\{(r,t)|r\in\mathcal{B},t\in\mathcal{B}\}} \lambda_1(\epsilon_{s,i} - \epsilon_{s,j} + y_i - y_j)(\hat{S}_i - \hat{S}_j)\nabla(\hat{S}_i - \hat{S}_j)$$
$$+\lambda_2(\epsilon_{S,i} - \epsilon_{S,j} + y_i - y_j)(\hat{S}_i - \hat{S}_j)\nabla(\hat{S}_i - \hat{S}_j) \tag{20}$$

$$\mathbf{g}^* = -\frac{1}{|\mathcal{B}|} \sum_{(i,j)\in\{(r,t)|r\in\mathcal{B},t\in\mathcal{B}\}} \lambda_1(y_i - y_j)(\hat{S}_i - \hat{S}_j)\nabla(\hat{S}_i - \hat{S}_j)$$

$$+ \lambda_2(y_i - y_j)(\hat{S}_i - \hat{S}_j)\nabla(\hat{S}_i - \hat{S}_j) \tag{21}$$

Thus, we should verify the original loss function without gradient noise (i.e. $\mathcal{L}_\beta^*$) satisfies the following assumptions.

**Verify Assumption 5.1 in (Bu et al., 2024).** Since $y_i \in [0,1]$, $\hat{S}_i \in [0,1]$, we have $\mathcal{L}_\beta^* \geq -n^2$. Thus, it has lower bound and satisfies Assumption 5.1 in (Bu et al., 2024).

**Verify Assumption 5.2 in (Bu et al., 2024).** According to Appendix. A.9, we can get the Hessian of $\mathcal{L}_\beta^*$ in Eq. 22, where $\nabla_{W_1,W_2}^2 \hat{S} = \nabla_{W_2,W_1}^2 \hat{S} = \hat{S}(1-\hat{S})\operatorname{diag}(\sigma_1')I_k \otimes \bar{S}^T$. $W_1 \in R^{k\times(D+2)}$, $W_2 \in R^{1\times r}$ and $I_k$ denotes a $k \times k$ identity matrix. $\sigma_1'$ is given in Eq. 23. $\otimes$ denotes the Kronecker product.

$$\frac{\partial^2 \mathcal{L}_\beta^*}{(\partial \mathcal{P})^2} = \begin{bmatrix} \frac{\partial^2 \mathcal{L}_\beta^*}{(\partial W_1)^2} & \frac{\partial^2 \mathcal{L}_\beta^*}{(\partial W_1)(\partial W_2)} \\ \frac{\partial^2 \mathcal{L}_\beta^*}{(\partial W_2)(\partial W_1)} & \frac{\partial^2 \mathcal{L}_\beta^*}{(\partial W_2)^2} \end{bmatrix}$$

$$= \begin{bmatrix} 0 & (y_i - y_j)(\nabla_{W_1,W_2}^2 \hat{S}_i - \nabla_{W_1,W_2}^2 \hat{S}_j) \\ (y_i - y_j)(\nabla_{W_2,W_1}^2 \hat{S}_i - \nabla_{W_2,W_2}^2 \hat{S}_j) & 0 \end{bmatrix} \tag{22}$$

$$\sigma_1'[i] = \begin{cases} k_1, & W_1\hat{S}[i] \geq 0 \\ k_2, & others \end{cases} \tag{23}$$

Since $\hat{S}$ only refers to the $s$, $S$ and data representation obtained from TSADM $R$, which are constants when training CoLLaTe (because the TSADM and LLM are fixed), $\bar{S}$ have a upper bound $U$ for a given dataset. Furthermore, since $\hat{S} \in [0,1]$, $\left|\nabla_{W_1,W_2}^2 \hat{S}\right| \leq \max(|k_1|, |k_2|) \times k^2 U$. Because $y_i \in [0,1]$ and $y_j \in [0,1]$, the Hessian matrix $\left|\frac{\partial^2 \mathcal{L}_\beta^*}{(\partial \mathcal{P})^2}\right| \leq 4\max(|k_1|, |k_2|) \times k^2 U$. Thus, $\mathcal{L}*_\beta$ is Lipschitz smooth and satisfies the Assumption 5.2 in (Bu et al., 2024).

**Verify Assumption 5.3 in (Bu et al., 2024).** We can transform $\mathbb{E}(\mathbf{g}_{i,j}^* - \mathbf{g}_{i,j})$ step by step, as shown in Eq. 24.

$$\mathbb{E}(\mathbf{g}_{i,j}^* - \mathbf{g}_{i,j}) = \mathbb{E}[(\lambda_1(\epsilon_{s,i} - \epsilon_{s,j}) + \lambda_2(\epsilon_{S,i} - \epsilon_{S,j}))(\hat{S}_i - \hat{S}_j)\nabla(\hat{S}_i - \hat{S}_j)]$$

$$= \mathbb{E}[(\lambda_1(\epsilon_{s,i} - \epsilon_{s,j}) + \lambda_2(\epsilon_{S,i} - \epsilon_{S,j}))](\hat{S}_i - \hat{S}_j)\nabla(\hat{S}_i - \hat{S}_j) \tag{24}$$

Since $\mathbb{E}(\epsilon_{s,i}) = \mathbb{E}(\epsilon_{s,j})$, $\mathbb{E}(\epsilon_{S,i}) = \mathbb{E}(\epsilon_{S,j})$, we can obtain $\mathbb{E}[\lambda_1(\epsilon_{s,i} - \epsilon_{s,j}) + \lambda_2(\epsilon_{S,i} - \epsilon_{S,j})] = 0$. Thus, $\mathbb{E}(\mathbf{g}_{i,j}^* - \mathbf{g}_{i,j}) = 0$. $\mathbb{E}|\mathbf{g}_{i,j}^* - \mathbf{g}_{i,j}|^2$ can be calculated as shown in Eq. 25.

$$\mathbb{E}|\mathbf{g}_{i,j}^* - \mathbf{g}_{i,j}|^2 = (\hat{S}_i - \hat{S}_j)^2(\nabla(\hat{S}_i - \hat{S}_j))^2\mathbb{E}[(\lambda_1(\epsilon_{s,i} - \epsilon_{s,j}) + \lambda_2(\epsilon_{S,i} - \epsilon_{S,j}))^2] \tag{25}$$

As proven above, $|\hat{S}_i - \hat{S}_j| \leq 1$, thus $(\hat{S}_i - \hat{S}_j)^2 \leq 1$. Similarly $\nabla(\hat{S}_i - \hat{S}_j) \leq 1$, we can obtain $(\nabla(\hat{S}_i - \hat{S}_j))^2 \leq 1$. Thus, we can obtain the inequality in Eq. 26.

$$(\hat{S}_i - \hat{S}_j)^2(\nabla(\hat{S}_i - \hat{S}_j))^2\mathbb{E}[(\lambda_1(\epsilon_{s,i} - \epsilon_{s,j}) + \lambda_2(\epsilon_{S,i} - \epsilon_{S,j}))^2] \leq$$

$$\mathbb{E}[(\lambda_1(\epsilon_{s,i} - \epsilon_{s,j}) + \lambda_2(\epsilon_{S,i} - \epsilon_{S,j}))^2] \tag{26}$$

Since $\epsilon_{s,i} \sim \mathcal{D}(\mu_s, \sigma_s)$, $\epsilon_{s,j} \sim \mathcal{D}(\mu_s, \sigma_s)$ independently, $\mathbb{E}[(\epsilon_{s,i} - \epsilon s, j)^2] = 2\sigma_s^2$. Similarly, $\mathbb{E}[(\epsilon_{S,i} - \epsilon_{S,j})^2] = 2\sigma_S^2$. After simplifying its formation, we obtain $\mathbb{E}[(\lambda_1(\epsilon_{s,i} - \epsilon_{s,j}) + \lambda_2(\epsilon_{S,i} - \epsilon_{S,j}))^2] = 2\lambda_1^2\sigma_s^2 + 2\lambda_2^2\sigma_S^2$. Thus, $\mathbb{E}|\mathbf{g}_{i,j}^* - \mathbf{g}_{i,j}|^2 \leq 2\lambda_1^2\sigma_s^2 + 2\lambda_2^2\sigma_S^2$. Then, it satisfies Assumption 5.3 in (Bu et al., 2024).

Therefore, $\mathcal{L}_\beta^*$ satisfies Assumption 5.1-Assumption 5.3 in (Bu et al., 2024). As proven before, minimizing $\mathcal{L}_\beta$ by stochastic gradient descent can be deemed as adding a random noise to the gradient of each sample during the stochastic gradient descent process of $\mathcal{L}_\beta^*$. Since Assumption 5.1-Assumption 5.3 in (Bu et al., 2024) hold, we can use Theorem 4 in (Bu et al., 2024) to prove that after adding these random noise, $\mathcal{L}_\beta$ can also converge as quickly as $\mathcal{L}_\beta^*$ by $\mathbf{O}(T^{-\frac{1}{4}})$. Besides, we have the equation as shown in Eq. 27.

$$\mathbb{E}(\mathbf{g}_{i,j}) = \mathbb{E}(\mathbf{g}_{i,j}^*) + \mathbb{E}([\lambda_1(\epsilon_{s,i} - \epsilon_{s,j}) + \lambda_2(\epsilon_{S,i} - \epsilon_{S,j})](\hat{S}_i - \hat{S}_j)\nabla(\hat{S}_i - \hat{S}_j)) \tag{27}$$

Since $\mathbb{E}[\lambda_1(\epsilon_{s,i} - \epsilon_{s,j}) + \lambda_2(\epsilon_{S,i} - \epsilon_{S,j})] = 0$, we can obtain $\mathbb{E}([\lambda_1(\epsilon_{s,i} - \epsilon_{s,j}) + \lambda_2(\epsilon_{S,i} - \epsilon_{S,j})](\hat{S}_i - \hat{S}_j)\nabla(\hat{S}_i - \hat{S}_j)) = 0$. Thus, $\mathbb{E}(\mathbf{g}_{i,j}) = \mathbb{E}(\mathbf{g}^*_{i,j})$. Since $\mathcal{L}_\beta$ can converge as proven above, its gradient $\mathbb{E}(\mathbf{g})$ should converge to 0 as iteration step grows up. Thus, as iteration step grows up, $\mathbb{E}(\mathbf{g}_{i,j})$ approaches 0, i.e., $\mathcal{P}$ approaches the one that makes $\mathbb{E}(\mathbf{g}^*_{i,j}) = 0$. Since $\mathcal{L}^*_\beta$ is convex, as long as $\mathbb{E}(\mathbf{g}^*_{i,j}) = 0$, $\mathcal{P}$ is the optimal solution of $\mathcal{L}^*_\beta$. Thus, by using $\mathcal{L}_\beta$, $\mathcal{P}$ can converge to the optimal solution of $\mathcal{L}^*_\beta$ with similar converge rate $\mathbf{O}(T^{-\frac{1}{4}})$, as iteration step approaches infinite.

## A.4 PROOF OF THEOREM 2

Since when Assumption 1 - Assumption 4 hold, Lemma 1 is valid. Thus, using stochastic gradient descent (SGD) to minimize $\mathcal{L}_\beta$ can be approximate to using stochastic gradient descent to minimize $\mathcal{L}^*_\beta$, when iteration step approach infinity. Thus, we prove that optimal solution of $\mathcal{L}^*_\beta$ has the mentioned properties.

*Proof of property 2.* Let $\hat{S}^\dagger$ denote the optimal solution of $\mathcal{L}^*_\beta$. We use the proof of contradiction to prove it. We firstly assume if there are $r, k, r', k' \in [1, n]$ that satisfies Eq. 28-Eq. 29. By enlarging $\hat{S}^\dagger_r$ by $\Delta t$ and reducing $\hat{S}^\dagger_{r'}$ by $\Delta t$, we can further decrease the value of $\mathcal{L}^*_\beta$, which is contradict to the assumption that $\hat{S}^\dagger$ is optimal. Thus, $\forall r, k, r', k'$, if $y_r - y_k > y_{r'} - y_{k'}$, then $\hat{S}^\dagger_r - \hat{S}^\dagger_k > \hat{S}^\dagger_{r'} - \hat{S}^\dagger_{k'}$.

Here is the detailed derivation. Without loss of generality, we can assume $y_r > y_{r'}$, because if $y_r < y_{r'}$, given $y_r - y_k < y_{r'} - y_{k'}$, $y_k < y_{k'}$. In this case, by multiplying minus one to both side of the inequality Eq. 28-Eq. 29, we can obtain a set of inequality that $y_{k'} - y_{r'} > y_k - y_r$, $\hat{S}^\dagger_{k'} - \hat{S}^\dagger_{r'} < \hat{S}^\dagger_k - \hat{S}^\dagger_r$ with the condition $y_{k'} > y_k$, which is equal to assuming $y_r > y_{r'}$ for Eq. 28-Eq. 29.

$$y_r - y_k > y_{r'} - y_{k'} \tag{28}$$

$$\hat{S}^\dagger_r - \hat{S}^\dagger_k < \hat{S}^\dagger_{r'} - \hat{S}^\dagger_{k'} \tag{29}$$

Let $\hat{S}^\dagger_r$ grows to $\hat{S}^\dagger_r + \Delta t$ and $\hat{S}^\dagger_{r'}$ decreases to $\hat{S}^\dagger_{r'} - \Delta t$, where $\Delta t > 0$. Let $\mathcal{L}_1$ denote the value of $\mathcal{L}^*_\beta$ before changing $\hat{S}^\dagger_r$ and $\hat{S}^\dagger_{r'}$, and $\mathcal{L}_2$ denote the value of $\mathcal{L}^*_\beta$ after changing $\hat{S}^\dagger_r$ and $\hat{S}^\dagger_{r'}$. We compute the difference between $\mathcal{L}_2$ and $\mathcal{L}_1$ in Eq. 30. Furthermore, we can simplify it to Eq. 31. Given $y_r > y_{[r']}$, $\mathcal{L}_1 - \mathcal{L}_2 > 0$, which means $\mathcal{L}_2$ is smaller than $\mathcal{L}_1$. However, since $\hat{S}^\dagger$ is the optimal solution of $\mathcal{L}^*_\beta$, there should be no value set of $\hat{S}$ that can make $\mathcal{L}^*_\beta$ smaller than $\mathcal{L}_1$. Thus, the conclusion $\mathcal{L}_2 < \mathcal{L}_1$ is contradict to the assumption that $\hat{S}^\dagger$ is optimal. Thus, $\forall r, k, r', k'$, if $y_r - y_k > y_{r'} - y_{k'}$, then $\hat{S}^\dagger_r - \hat{S}^\dagger_k > \hat{S}^\dagger_{r'} - \hat{S}^\dagger_{k'}$.

$$\mathcal{L}_1 - \mathcal{L}_2 = \sum_{i=1, i\neq r}^n (y_i - y_{r'})\Delta t + \sum_{j=1, j\neq r} (y_{r'} - y_j)(-\Delta t) + (y_r - y_{r'})(2\Delta t) + (y_{r'} - y_r)(-2\Delta t)$$
$$+ \sum_{i=1, i\neq r'}^n (y_i - y_r)(-\Delta t) + \sum_{j=1, j\neq r'}^n (y_r - y_j)(\Delta t) \tag{30}$$

$$\mathcal{L}_1 - \mathcal{L}_2 = 2n\Delta t(y_r - y_{r'}) \tag{31}$$

*Proof of Property 1.* Let $k = k'$. Then, according to property 2, $\forall r, k, r'$, if $y_r - y_k > y_{r'} - y_k$, $\hat{S}^\dagger_r - \hat{S}^\dagger_k > \hat{S}^\dagger_{r'} - \hat{S}^\dagger_k$. The inequality mentioned above can further reduced to $\forall r, r'$, if $y_r > y_{r'}$, $\hat{S}^\dagger_r > \hat{S}^\dagger_{r'}$.

## A.5 TRAIN AND INFERENCE

During the training process, we use $\mathcal{L}_a + \mathcal{L}_\beta$ as the loss function. During the inference process, we use the collated anomaly score output by the conditional network as the anomaly score.

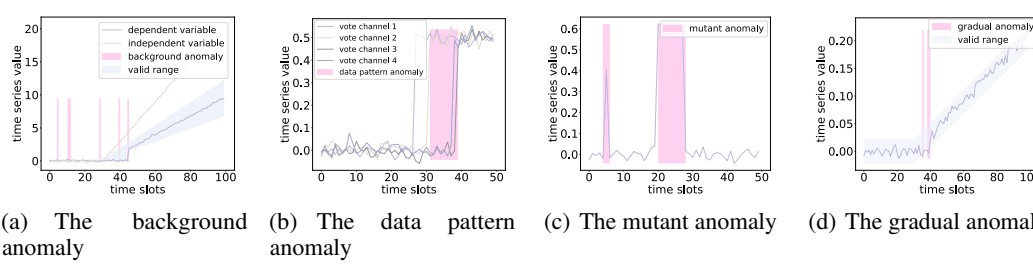

(a) The background anomaly     (b) The data pattern anomaly     (c) The mutant anomaly     (d) The gradual anomaly

Figure 4: These figures show different anomaly types

## A.6 ANOMALY TYPES

We use different binary classification standards to categorize the anomalies in Flight 1 and Flight 2. Among them, the most common binary standard is point anomaly and contextual anomaly, which is illustrated in (Lai et al., 2024). Besides, we also introduce two other kinds of binary classification standards for aircraft monitoring datasets to explore the complementary performance between LLM and TSADM. According to the degree of reliance on expert knowledge for anomaly detection, we can divide anomalies into background anomalies and data pattern anomalies. The background anomalies highly rely on expertise knowledge, while the data pattern anomaly can be identified only by TSADMs. As shown in Fig. 4(a), the valid range of the dependent variable is decided by the independent variable, which should satisfy the laws of physics between different parameters. It is usually hard for a neural network to clearly identify the valid range of the dependent variable, given the independent variable. Thus, in this case, it relies on expert knowledge and belongs to background anomaly. In contrast, in Fig. 4(b), the redundant channel mechanism usually requires that most of the redundant channels can reach a consensus. Thus, different redundant channels should always be similar to each other in normal status. In this situation, the neural network can identify the similarity between different channels without expert knowledge. Thus, it belongs to data pattern anomaly. According to the similarity between anomaly and surrounding time series, we can divide anomalies into gradual anomalies and mutant anomalies. As shown in Fig. 4(c), the anomalies are quite different from surrounding normal time series and they should belong to mutant anomalies. In Fig. 4(d), the anomalies are similar to the surrounding normal time series and they should belong to the gradual anomaly.

## A.7 DATASETS

In our experiments, we used four datasets from different scenarios: cloud service monitoring, aircraft monitoring, and mathematical assumption-based anomaly datasets. We introduce these datasets in the following. The used datasets of Mackey and Mustang are included in "Supplementary Material". As for Flight 1 and Flight 2, since they refers to business secret, we could not disclose them.

- **Mackey** (Thill & Konen, 2020).The dataset comprises a synthetic Mackey-Glass time series characterized by non-trivial anomalies. Mackey-Glass time series are recognized for their propensity to exhibit chaotic behavior under specific conditions. The Mackey dataset encompasses 10 Megabytes (MG) of time series data, with each series extending to a length of 105 time points. Within each time series, 10 anomalies are meticulously introduced using a procedure outlined subsequently. Unlike other synthetic benchmarks, it poses a considerable challenge for the human eye to discern the artificially introduced anomalies from the inherent chaotic behavior. A segment of a time series featuring 3 anomalies is illustrated in the graph above. Due to the limit of computation resources when using LLM, we only use 10000 time slots in our experiments. Since there are no point anomalies in the original dataset, but our model utilizes the complementary performance of LLM and TSADM on point anomaly and contextual anomaly, we insert some point anomalies into the original dataset and release it in "Supplementary Material".
- **Mustang** (Amvrosiadis et al., 2018; Chen et al., 2024b). The Mustang dataset chronicles the task duration of operations over a span of five years. The authors in (Chen et al., 2024b) have conducted preprocessing on the raw dataset, where they count the histogram of the task duration time distribution at each time slot. Subsequently, they manually identified and labeled the slowdown anomalies. We include used dataset in "Supplementary Material".

Table 4: The meaning of each hyperparameter.

| Hyperparameter Name | Meaning |
|---|---|
| winLen | The length of sliding window |
| moduleNum | The number of TSADM attention layers |
| batchSize | The size of batch |
| trlr | The learning rate of TSADM |
| colr | The learning rate during collaboration |
| patchSize | The size of patch in computin $D_{inter}$ and $D_{intra}$ |
| kLen | The length of convolution kernel in TSADM |
| $d$ | The square root in Eq. 37 |
| $\hat{\lambda}$ | The hyperparameter weight in $\mathcal{L}_a$ |

Table 5: The value of each hyperparameter.

| Mustang | | Mackey | | Fligh 1 | | Flight 2 | |
|---|---|---|---|---|---|---|---|
| Name | Value | Name | Value | Name | Value | Name | Value |
| winLen | 4 | winLen | 5 | winLen | 2 | winLen | 2 |
| moduleNum | 6 | moduleNum | 10 | moduleNum | 6 | moduleNum | 10 |
| batchSize | 100 | batchSize | 100 | batchSize | 100 | batchSize | 100 |
| trlr | 0.001 | trlr | 0.01 | trlr | 0.01 | trlr | 0.01 |
| colr | 0.001,0.0001 | colr | 0.01 & 0.001 | colr | 0.008 | colr | 0.01,0.001 |
| patchSize | 2 | patchSize | 2 | patchSize | 2 | patchSize | 2 |
| kLen | 2 | kLen | 2 | kLen | 2 | kLen | 2 |
| d | 0.7,1.2 | d | 0,0.5,0.7,1,1.5 | d | 0,1.7,2 | d | 0.01,0.5,0.9,1.7,2 |
| $\hat{\lambda}$ | 0.1,1 | $\hat{\lambda}$ | 1 | $\hat{\lambda}$ | 1,2,5 | $\hat{\lambda}$ | 1,5 |

- **Flight 1 & 2**. These datasets are collected from an airline company, which is one of the world's top 150 enterprises. Due to the confidentiality requirements of monitoring data, we extracted obvious data patterns from the raw data and performed encryption and desensitization operations on the raw data through regeneration. The datasets use redundant channels (Osder, 1999) to ensure the robustness of flight systems and refer to sophisticated collaborative relationships between different parameters, which introduces complex rules to judge the availability of the flight control system and is hard to learn only by neural networks. Besides, we show the ratios of normality and different kinds of anomaly in Fig. 5(a), where the anomaly classification is defined in Appendix. A.6.

### A.8 EXPERIMENT DETAILS

The four datasets used in our experiments consist of multiple subsets. For each subset, we use 40% as the training set, 10% as the validation set, and the remaining 50% as the testing set. We illustrate the meaning of each hyperparameter in Tab. 4 and list the value of the hyperparameter in Tab. 5. Since the datasets used in our experiments consist of several subsets. In some datasets, we set different hyperparameter values for different subsets and we list the values used in the Tab. 5. In these situations, we use grid search to find the optimal combinations of these hyperparameters for each subset.

### A.9 ARCHITECTURE OF OUR CONDITIONAL NETWORK

The conditional network uses the data representation obtained from a TSADM as the condition and leverages the aligned anomaly score of the TSADM and anomaly score of LLM to output the collated anomaly score. As shown in Eq. 32, we firstly concat data representation, aligned anomaly score of TSADM and anomaly score of LLM as many conditional networks do (Rakelly et al., 2018). After that, we use multilayer perceptrons to output collated anomaly score as shown in Eq. 33, where $W_1$ and $W_2$ are trainable parameters, $\sigma_1$ is LeakyReLU and $\sigma_2$ is Sigmoid.

$$\bar{S} = \text{Concat}(S, s, R) \tag{32}$$

$$\hat{S} = \sigma_2(W_2\sigma_1(W_1\bar{S})) \tag{33}$$

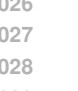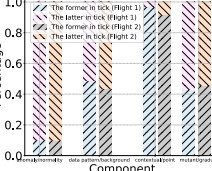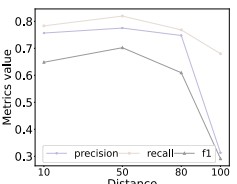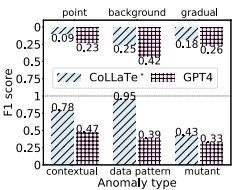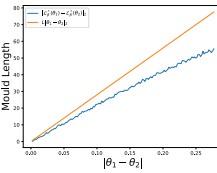

(a) The distribution of Flight datasets

(b) GPT4 performance for different distances from examples

(c) The complementary strength of LLM and TSADM

(d) Validity of Assumption 3

Figure 5: (a) shows the distribution of flight datasets. (b) shows the anomaly detection performance of GPT4 when the distance between the designated time slot for anomaly detection and the time slot of the examples increases. (c) The figure shows the F1 score of LLM (GPT4) and TSADM (CoLLaTe$^\star$) when using different anomaly binary classification criteria. (d) We verify that the mould length of $|\mathcal{L}_\beta^*(\theta_1) - \mathcal{L}_\beta^*(\theta_2)|_2$ is less than $L|\theta_1 - \theta_2|_2$ and verifies the validity of assumption 3.

## A.10 ARCHITECTURE OF TSADM

The TSADM consists of 6-layer anomaly attention. Each anomaly attention layer is described in Eq. 34, where $G[i, j] = 1 - \exp^{-\frac{(i-j)^2}{\sigma^2}}$, $\sigma$ is a learnable parameter and $*$ denotes an element-wise multiplication. This attention mechanism is built on the observation that anomaly could not establish decentralized connection with the whole time series (Xu et al., 2022) and embeds moderate modification compared with the original one. To promote the anomaly attention capacity of dealing with compound periodic time series, different anomaly attention layers are organized by skimming attention mechanism (Chen et al., 2024b).

$$\mathcal{A} = q_d k_d^T, \tag{34}$$

$$\tilde{\mathcal{A}} = \mathrm{softmax}(\mathcal{A} * G), \tag{35}$$

$$\tilde{x}_l[:, d] = \tilde{\mathcal{A}} v_d. \tag{36}$$

## A.11 COMPLEMENTARY PERFORMANCE

In CoLLaTe, we mainly utilize the complementary performance of LLM and TSADM on point anomaly and contextual anomaly. Actually, there are other complementary performances between the TSADM and LLM by using another binary classification standard, and we explore them in this section. We introduce two kinds of binary classification standards: the anomalies that need expert knowledge to detect and the anomalies that can be detected only by data patterns, gradual anomalies, and mutual anomalies. These anomalies are illustrated in detail in Appendix. A.6. We divide Flight 2 into pieces and gather the pieces only obtain only one kind of anomaly. We test GPT4 and CoLLaTe$^\star$ on them and compute their average F1 score, which is shown in Fig. 5(c). In Fig. 5(c), we can see that LLM always has better performance on one kind of anomaly and has poorer performance on another in each binary anomaly classification.

## A.12 SET-UP-PITCH PROMPT

Numerous studies have shown that incorporating similar historical examples into prompts can significantly enhance anomaly detection performance (Liu et al., 2024b). Based on this insight, our prompt is structured into four components: expertise supplementation, task description, input data, and examples. The expertise supplementation component provides the contextual meanings of the various dimensions of the input data within the application domain, along with relevant domain-specific knowledge in professional document, such as typical value ranges and interrelationships between dimensions. The task description component directs the large language model (LLM) to determine whether a given time slot is anomalous and specifies the required output format. The examples component includes both a positive and a negative example to guide the model's understanding. The detailed prompt design is provided in the following.

Table 6: The mean and standard deviation of CoLLaTe performance.

| | Mustang | | | Mackey | | | Flight1 | | | Flight2 | | |
|---|---|---|---|---|---|---|---|---|---|---|---|---|
| | Pre | Rec | F1 | Pre | Rec | F1 | Pre | Rec | F1 | Pre | Rec | F1 |
| Best Baseline | **1.000** | 0.891 | 0.935 | 0.829 | 0.960 | 0.882 | 0.690 | 0.947 | 0.768 | 0.749 | 0.742 | 0.746 |
| CoLLaTe-mean | 0.934 | **0.989** | **0.956** | **0.941** | **0.990** | **0.964** | **0.806** | **0.970** | **0.872** | **0.841** | **0.930** | **0.871** |
| CoLLaTe-std | 0.044 | 0.025 | 0.015 | 0.023 | 0.013 | 0.017 | 0.025 | 0.019 | 0.010 | 0.068 | 0.040 | 0.026 |

While recent works utilize examples from a fixed dataset, many application scenarios exhibit dynamic patterns over time. For instance, in cloud server monitoring, the cloud environment evolves with service updates, deployments, and revocations (Chen et al., 2024a; Ma et al., 2021). Similarly, in aircraft monitoring, normal patterns shift with changes in flight conditions and aircraft states. As illustrated in Fig. 5(b), our experiments with aircraft monitoring data reveal that the performance of GPT-4 degrades as the temporal distance between the anomaly detection time slot and the example time slot increases. To address this, we leverage a TSADM for anomaly detection to periodically label data flow and update the example collection with manual assistance, ensuring that the examples remain relevant and effective over time.

We list the concrete prompts for different datasets below. Since the prompt for aircraft monitoring datasets (Flight 1 and Flight 2) refers to the trade secrets of the airline company, we do not list them below. Due to the randomness of LLM, we ran 5 times independently and show the average performance of CoLLaTe and its standard deviation in Table. 6.

---

**Prompt of Mackey**
**Expertise Supplement**: The input data is a $10000 * 1$ time series. The time series is generated by adding the value x, which satisfied $\frac{dx}{dt} = 0.25 * \frac{x(t-18)}{1+x(t-18)^{10}} - 0.1 * x(t)$, and some noises within range of $[-0.01, 0.01]$, where $x(t)$ represents the value of time series at $t^{th}$ time slot. There are some anomalies inserted into the time series, which do not obey mentioned rules. The anomalies are inserted by repeating some future segments of the time series to the present position. [Professional document can be inserted into this part]
**Input data**: *{data}*
**Task description**: Given the input data, please pick out the anomalies along the time series and output the possibility that the $i^{th}$ time slot is anomalous. Please only output the corresponding possibility within the range of $[0, 1]$. ($i$ is an iterable index).
**Examples**:
Example 1: Given the input data, please pick out the anomalies along the time series and output the possibility that the $i^{th}$ time slot is anomalous. Please only output the corresponding possibility within the range of $[0, 1]$. Output: 0.

Example 2: Given the input data, please pick out the anomalies along the time series and output the possibility that the $i^{th}$ time slot is anomalous. Please only output the corresponding possibility within the range of $[0, 1]$. Output: 1.

---

**Prompt of Mustang**
**Expertise Supplement**: The input data is a matrix with size T * 17, which represents the task duration time distribution for T time slots in the cloud center. For each time slot, there are 17 dimensions. The values for these dimensions respectively represent the ratio of tasks whose duration time belongs to [0,5), [5,10), [10, 20), [20,30), [30, 40), [40, 70), [70, 110), [110, 150), [150, 190), [190, 230), [230, 280), [280, 330), [330, 380), [380, 430), [430, 900), [900, 1200), [1200, 1900). [Professional document can be inserted into this part]
**Input data**: *{data}*.
**Task description**: Given a sliding window with size $50 * 17$, please judge if there are many tasks slowdown at a specific time slot, compared with other slots in a given window. Please output a float number ranging from 0 to 1 to represent the probability that there are many tasks slowdown at the specific time slot.

---

Table 7: The model efficiency.

| Model | Train (s/subset) | Inference(s/subset) | LLM (s/request) |
|---|---|---|---|
| CoLLaTe | 12.27 | 0.24 | 1.27 |
| GPT4 | — | — | 1.27 |
| sigLLM | — | — | 2.31 |
| LLMAD | — | — | 1.42 |
| MSCRED | 3.04 | 0.17 | — |
| OmniAnomaly | 2.32 | 0.14 | — |
| TranAD | 1.52 | 1.31 | — |
| AnomalyTr | 9.46 | 1.71 | — |
| DCdetector | 3.79 | 1.21 | — |

**Examples**:

Example 1: Given the input data, please judge if there are many tasks that slow down at time slot $i$ and output a float number ranging from 0 to 1 to represent the probability of it. Response: 0.

Example 2: Given the input data, please judge if there are many tasks that slow down at time slot $i$ and output a float number ranging from 0 to 1 to represent the probability of it. Response: 1

## A.13 MIN-MAX NORMALIZATION TO TSADM ANOMALY SCORES

we firstly scale anomaly scores from the TSADM by Eq. 37, where $d$ is a hyperparameter, $\dot{s}_{max}$ and $\dot{s}_{min}$ are the maximum and minimum values of these anomaly scores.

$$s = \frac{\dot{s}}{\sqrt[d]{\dot{s}_{max} - \dot{s}_{min}}} \tag{37}$$

## A.14 MODEL EFFICIENCY

Since we call the GPT-4 API, which will force the program to sleep for a while after sending several requests consecutively. Therefore, using a locally deployable LLM would significantly reduce the time compared to the times we have listed.

When using a K80 to train and inference CoLLaTe (LLM part calls API), the average time to obtain LLM anomaly scores is 1.27 seconds per request for the first 10 requests. After obtaining the anomaly scores, we separately compute the training and inference times. Training CoLLaTe on the Mackey dataset takes an average of 12.27 seconds per subset, while inference takes an average of 0.24 seconds per subset. We compare the running time of CoLLaTe and the baselines in detail in Table 7, where the time for LLM is calculated separately from the time for other steps. The memory size of the CoLLaTe (except the LLM part) is 43KB.

Regarding time complexity, the time complexity for transformer-based methods (LLM and TSADM in CoLLaTe) is $O(n^2)$, where $n$ is the sequence length of prompt. The time complexity for the conditional network is $O(d^2)$, where $d$ is the dimension of the feedforward network in the conditional network. Thus, the total time complexity of CoLLaTe is $O(n^2 + d^2)$.

## A.15 POTENTIAL LIMITATION

Firstly, CoLLaTe is not suitable for application scenarios with limited computational capacity or unstable network accessibility, as we would be unable to deploy the LLM locally or call its API remotely. Secondly, CoLLaTe is not applicable in scenarios with high concurrency or stringent real-time requirements, as calling the GPT-4 API takes 1 second per request and triggers forced delays after several consecutive requests. Besides, if both LLM and TSADM in CoLLaTe fail to detect an anomaly, CoLLaTe will also fail to detect it.

## A.16   ASSUMPTION VALIDITY

Assumptions 1 and 2 state that the anomaly score prediction errors of LLM and TSADM follow two unknown distributions, respectively. The only constraint on these distributions is that their expected values are not equal to zero. We consider this assumption to be practical, as no model can achieve perfect performance, and the expectation of their prediction errors should naturally bigger than zero. This is also proven by the experimental result shown in Tab. 2, where the F1 scores of CoLLaTe$^\star$ and GPT4 are not 1.0. Assumption 3 assumes that $\mathcal{L}_\beta^*$ in Eq. 6 is $L$-Lipschitz continuous towards the trainable parameters in $\hat{S}_i$ and $\hat{S}_j$. We plot the variation of $|\mathcal{L}_\beta^*(\theta_1) - \mathcal{L}_\beta^*(\theta_2)|$ as the distance between two trainable parameters $\theta_1$ and $\theta_2$ change in Fig. 5(d). We can assure $|\mathcal{L}_\beta^*(\theta_1) - \mathcal{L}_\beta^*(\theta_2)| < L|\theta_1 - \theta_2|$ when setting $L$ as 280 and the Lipschitz continuous assumption is satisfied.

## A.17   EFFECTIVENESS OF EXPERT KNOWLEDGE

To verify the effectiveness of Expert knowledge used in prompt. We omit the expert knowledge and compare the original model performance with omitting one in Tab. 8, where 'w/o EK' denotes omit expert knowledge.

|  | Mustang | | | Mackey | | | Flight1 | | | Flight2 | | |
| --- | --- | --- | --- | --- | --- | --- | --- | --- | --- | --- | --- | --- |
|  | Prec | Rec | F1 | Prec | Rec | F1 | Prec | Rec | F1 | Prec | Rec | F1 |
| w/o EK | 0.701 | 0.936 | 0.801 | 0.810 | 0.978 | 0.883 | 0.699 | 0.828 | 0.756 | 0.618 | 0.936 | 0.744 |
| CoLLaTe | **1.000** | 0.943 | 0.967 | **0.970** | **0.996** | **0.982** | **0.790** | **0.978** | **0.866** | **0.897** | 0.891 | **0.883** |

Table 8: Expert Knowledge Ablation Study

## A.18   COMPARING WITH TIME SERIES FOUNDATION MODEL AND LLM

We compare CoLLaTe with Time Series Foundation Model, TimesNet (Wu et al., 2023), Timer (Liu et al., 2024c), Moment (Goswami et al., 2024), large language model, Qwen3 and popular time series models, FedFormer (Zhou et al., 2022), AutoFormer (Wu et al., 2021) in Table 9.

|  | Mustang | | | Mackey | | | Flight1 | | | Flight2 | | |
| --- | --- | --- | --- | --- | --- | --- | --- | --- | --- | --- | --- | --- |
|  | Prec | Rec | F1 | Prec | Rec | F1 | Prec | Rec | F1 | Prec | Rec | F1 |
| TimesNet | 0.448 | 0.317 | 0.371 | 0.647 | 0.324 | 0.431 | 0.904 | 0.765 | 0.829 | 0.701 | 0.747 | 0.723 |
| AutoFormer | 0.778 | **1.000** | 0.875 | 0.607 | 0.872 | 0.716 | 0.447 | 0.654 | 0.531 | 0.778 | 0.538 | 0.636 |
| FedFormer | 0.277 | 0.650 | 0.389 | 0.943 | **1.000** | 0.971 | **1.000** | 0.333 | 0.500 | 0.864 | 0.487 | 0.623 |
| CoLLaTe | **1.000** | 0.943 | **0.967** | **0.970** | 0.996 | **0.982** | 0.790 | 0.978 | **0.866** | **0.897** | 0.891 | **0.883** |

Table 9: Performance comparison of different methods on Mustang, Mackey, Flight1, and Flight2 datasets.

## A.19   USAGE OF LLM

In our paper writing process, we leverage large language models (LLMs) to assist with polishing and proofreading our English expressions and generating LaTeX tables from experiment results.

## A.20   ARCHITECTURE OF MAPPING FUNCTION

The Mapping function is designed as Eq. 38, where $R$ is time series representation obtained from a fixed TSADM. $\mathcal{M}(s_i)$ is a linear mapping function combined with a sigmoid function. Because both of them are monotonous and the time series representation part $R$ is a fixed constant (we do not change the learnable parameters of the pre-trained TSADM in our model), the $\mathcal{M}(s_i)$ is monotonous with respect to $s_i$.

$$\bar{s}_i = \text{Concat}(R, s_i), \mathcal{M}(s_i) = \text{Sigmoid}(W\bar{s}_i) \tag{38}$$

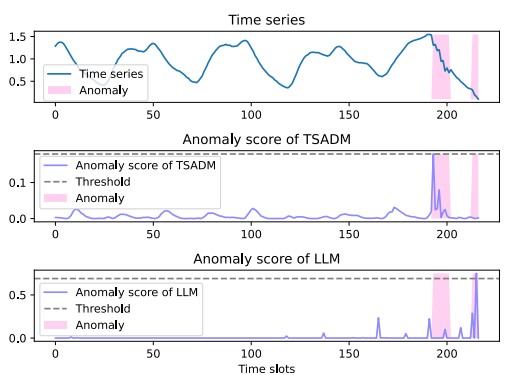 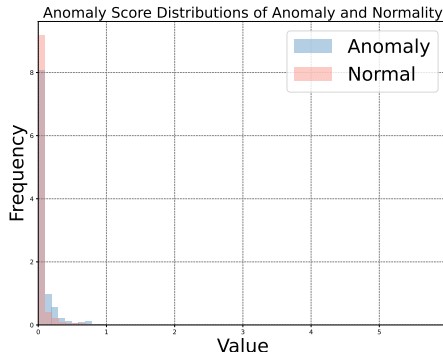

(a) The anomaly scores of LLM and TSADM on a time series

(b) The anomaly score distributions of anomaly and normality

Figure 6: (a) shows the anomaly scores of an LLM and a TSADM on different anomaly types; (b) shows the anomaly score distributions of normality and anomaly for TSFM.

## A.21 Where does LLM help?

| Distributions | D | Bootstrap-P-Value |
|---|---|---|
| Beta | 0.2023 | 0.5033 |
| Logit-normal | 0.2519 | 0.5633 |
| Half-Gaussian | 0.2428 | 0.4867 |
| GMM | 0.1218 | 0.4967 |

Table 10: Goodness-of-fit results for different distributions.

Table 11: Performance (Precision, Recall, F1) across four datasets.

| Model | Mustang | | | Mackey | | | Flight1 | | | Flight2 | | |
|---|---|---|---|---|---|---|---|---|---|---|---|---|
| | Pre | Rec | F1 | Pre | Rec | F1 | Pre | Rec | F1 | Pre | Rec | F1 |
| Beta | 0.933 | 0.992 | 0.961 | 0.851 | 0.992 | 0.915 | 0.792 | 0.972 | 0.860 | **0.965** | 0.848 | 0.885 |
| Logit-normal | 0.875 | **1.000** | 0.929 | 0.926 | **0.996** | 0.956 | **0.987** | 0.779 | 0.847 | 0.780 | **1.000** | 0.876 |
| Half-Gaussian | **1.000** | 0.943 | 0.967 | **0.970** | **0.996** | **0.982** | 0.790 | **0.978** | 0.866 | 0.897 | 0.891 | 0.883 |
| GMM | 0.980 | **1.000** | **0.989** | 0.962 | **0.996** | 0.978 | 0.805 | 0.978 | **0.872** | 0.835 | 0.978 | **0.889** |

We show a typical case where LLM can help and where it can not in Figure. 6(a). The first anomaly (before the 200-th time step) is a contextual anomaly: its absolute values appear normal when viewed globally, but they violate the expected local context distribution. The second anomaly is a point anomaly, because the global value range of the series lies above 2.66, while this spike falls below that range. For the first anomaly, the LLM fails to detect it because it is less sensitive to fine-grained fluctuations and local temporal patterns, whereas the TSADM is better at capturing such patterns and therefore detects it successfully. In contrast, the second anomaly requires global-range awareness to identify its abnormality. In this case, the LLM provides stronger global reasoning than the TSADM and thus performs better.

Table 12: Performance comparison across datasets for TSADM, LLM, ensemble methods, and CoLLaTe.

| | Mustang | | | Mackey | | | Flight1 | | | Flight2 | | |
|---|---|---|---|---|---|---|---|---|---|---|---|---|
| | Pre | Rec | F1 | Pre | Rec | F1 | Pre | Rec | F1 | Pre | Rec | F1 |
| TSADM → LLM | 0.696 | 0.889 | 0.571 | 0.259 | **1.000** | 0.411 | 0.375 | 0.857 | 0.522 | 0.416 | 0.877 | 0.458 |
| LLM → TSADM | 0.900 | 0.927 | 0.889 | 0.918 | 0.787 | 0.820 | 0.793 | 0.819 | 0.706 | 0.667 | 0.923 | 0.774 |
| Max | 0.857 | 0.861 | 0.845 | 0.768 | **1.000** | 0.863 | 0.845 | 0.855 | 0.794 | 0.778 | 0.824 | 0.799 |
| Avg | 0.874 | 0.907 | 0.881 | 0.812 | **1.000** | 0.891 | **0.898** | 0.734 | 0.776 | 0.563 | **1.000** | 0.720 |
| Vote | 0.076 | **1.000** | 0.141 | 0.177 | **1.000** | 0.301 | 0.157 | 0.889 | 0.267 | 0.112 | 0.876 | 0.199 |
| CoLLaTe | **1.000** | 0.943 | **0.967** | **0.970** | 0.996 | **0.982** | 0.790 | **0.978** | **0.866** | **0.897** | 0.891 | **0.883** |

| | Mustang | Mackey | Flight1 | Flight2 |
|---|---|---|---|---|
| $T_{\text{CoLLaTe}} - T_{\text{w/o-LLM}}$ (s/sample) | 1.31 | 1.24 | 1.26 | 1.41 |
| $F1_{\text{CoLLaTe}} - F1_{\text{w/o-LLM}}$ (%) | 2.9 | 2.5 | 17.2 | 18.1 |

Table 13: Performance and runtime improvements of CoLLaTe over the w/o-LLM variant.

### A.22 WHY TSFM SHOWS WEAK PERFORMANCE IN OUR EXPERIMENT?

The weak performance of TSFM can come from the overgeneralization problem in anomaly detection. Those TSFM are used as prediction-based anomaly detection method in our experiment. Since they are trained on lots of dataset in pretraining process, it has good generalization across different data patterns, which is verified by the zero-shot prediction performance reported by many TSFM papers [1,2]. However, this strength in zero-shot prediction can be a shortage when using it as a prediction-based anomaly detection, because it can also fit the anomaly patterns well. In such situation, the prediction error (used as anomaly score) of anomalies can also be small and is hard to distinguish from normal samples. We draw the anomaly score distributions of anomaly and normal patterns for TSFM in Figure. 6(b), where their distributions are mostly overlapped.

### A.23 GOODNESS-OF-FITS TEST

(Also guided by Reviewer 2vmL and Reviewer ct8Q). We provide goodness-of-fits test (KS) across half-Gaussian/Beta/logit-normal/isotonic calibration in Table. 10. According to the goodness-of-fits test, GMM can fit the distribution best.

We rerun our model with Beta distribution, Logit-normal and GMM, and list their performance in Table. 11. Since GMM can fit the LLM anomaly score distribution better, it can achieve better performance in most of datasets. Besides, our model is also robust for other assumptions.

### A.24 MORE BASELINES

We list the performance of TSADM -> LLM, LLM -> TSADM and simple score fusion strategies in Table. 12, where CoLLaTe still achieve the best F1 score.

- LLM -> TSADM: We first train a TSADM on a given dataset. Then, we fine-tune the TSADM with the anomaly scores of an LLM by distillation learning.

- TSADM -> LLM: We integrate the anomaly score output by a TSADM with the original LLM prompt.

- Max/Avg: We compute the Max/Avg of the anomaly scores from a TSADM and a LLM.

- Vote: We use a TSADM and two randomly sampled LLM output to vote.

### A.25 DISCUSSION OF COSTS VS. GAINS

we compare the time overhead of CoLLaTe and CoLLaTe-w/o LLM, as well as their F1 score in Table. 13. Since the Mackey dataset is based on the Mackey-glass equation and the Mustang dataset is to detect cluster-wide slow down from task execution time distribution, detecting anomalies from them depends more on the data fluctuation patterns rather than expert knowledge. As for Flight 1 and Flight 2 are monitoring data collected from aircraft, detecting anomaly from them depends on prior knowledge about aircraft design. Thus, incorporating LLM can improve F1 score on Flight 1 and Flight 2.

Considering the strength and weakness of CoLLaTe, it is better suited for situations where accuracy matters more than strict real-time response and require lots of expert knowledge to detect anomaly. For example, during ground tests after aircraft assembly, engineers prefer spending extra time to ensure the aircraft's health is assessed correctly, because a missed anomaly could lead to serious accidents during aircraft flight tests, which can lead to a plane crash and people death. Similar situations include safety checks before restarting large industrial machines, inspections of power systems before they go online, or pre-mission checks for UAVs and ships. In all these cases, taking a little longer is acceptable, but making a wrong decision is not.

### A.26 INTERPRETABILITY OF COLLATE

We can obtain an interpretation of CoLLaTe by the following step:

- Compute $D_{inter}$ and $D_{intra}$ and see which model contributes more to the final result.
- If the TSADM contributes more to the final result: we can compute reconstruction error at each dimension and localize the root cause of the anomaly.
- If LLM contributes more to the final result: we can get interpretation from LLM's response by removing the requirement "only output the anomaly score". Then, we can get the chain of thought of LLM.

We provide an example of LLM's response in the following:

---

**LLM API Response**

**LLM Output:** To determine if there are many tasks slow down at time slot 0 compared to the other time slots in the given sliding window, we can analyze the ratios of tasks for each time slot. Here's the breakdown for time slot 0 based on the given input data: - Time slot 0: [0.75, 0.08299999684095383, 0.16699999570846558, ...] - The first value (0.75) indicates that 75% of tasks have a duration in the range [0, 5). Now, we can compare this value to the other time slots to assess the overall distribution of tasks: From the first five time slots (for example): - Time slot 1: [0.4000000059604645, ...] - Time slot 2: [0.5, ...] - Time slot 3: [1.0, ...] - Time slot 4: [0.0, ...] In observing the first dimension, a value of 0.75 in time slot 0 is relatively high compared to slots 1, 2, and 4. However, time slot 3 has a value of 1.0, which indicates all tasks are in the [0, 5) duration. One approach to calculate a probability score for slot 0 is to evaluate the distribution of tasks across all time slots considering an anomaly detection score. Given there are examples with specific scores (0.000000 for time slot 0 indicating no slowdown) and 1.000000 for time slot 3 indicating a significant slowdown, we can infer that: - A score approaching 0 indicates normal conditions. - A score significantly higher than 0 indicates detected anomalies or slowdowns. Considering the distribution: - Time slot 0 has a rather high concentration of tasks with fast durations, while other time slots show varied distributions. Thus, the probability that time slot 0 has many tasks slow down compared to others can be assigned a score consistent with lower than the typical anomaly score. Based on the comparative analysis, I would assign a score of approximately 0.1 for time slot 0's probability of having many tasks slowed down compared to the overall data: Final output: Anomaly Score: 0.1

---

