# OpenReview forum: "Synergizing Large Language Models and Task-specific Models for Time Series Anomaly Detection"
_ICLR.cc/2026/Conference — Submitted to ICLR 2026_

### Official Review · Reviewer_ct8Q · 2025-10-28

**Soundness:** 3
**Presentation:** 2
**Contribution:** 3
**Rating:** 4
**Confidence:** 3

**Summary:**

The paper proposes CoLLaTe, a framework that integrates a task-specific time-series anomaly detector (TSADM) with a large language model (LLM) for enhanced anomaly detection. The core idea is to combine the numerical pattern recognition strength of TSADMs with the domain knowledge reasoning capability of LLMs. CoLLaTe introduces two main components:
(i) an alignment module that harmonizes the anomaly-score “expression domains” of the two models by fitting a half-Gaussian density to the LLM’s score distribution and learning a mapping that makes the TSADM scores follow this distribution; and
(ii) a fusion module that combines both scores through a conditional network trained with a collaborative loss, designed to mitigate the “error accumulation” observed with MSE-based objectives.

The paper provides theoretical analysis of the proposed loss and presents empirical results on four datasets, showing strong F1-score improvements and robustness to distribution shifts. Ablation studies further validate the contributions of each module.

**Strengths:**

S1: The paper’s identification of two key integration challenges, score misalignment and error accumulation are is both insightful and practically relevant for fusing heterogeneous detectors such as LLMs and TSADMs. The qualitative evidence in Fig. 1(b–d) (p. 3) effectively illustrates the semantic gap between their scoring behaviors and motivates the proposed alignment mechanism.

S2: The proposed architecture (Fig. 2, p. 3) is lightweight and modular, using a small conditional MLP that can be flexibly applied to various TSADM–LLM pairs. The reported memory footprint and the manageable training/inference latency demonstrate strong engineering efficiency and potential for real-world integration.

S3: The evaluation covers four datasets and includes module ablations, hyperparameter sensitivity analyses, and distribution-shift tests, consistently showing improved performance. This breadth of experimentation strengthens confidence in the method’s robustness and generality.

S4: The discussion on how MSE objectives can propagate bias when both sources are imperfect—and how the proposed pairwise collaborative loss mitigates such accumulation—is well reasoned. The theoretical properties, such as preserving relative ordering differences (Theorem 2), add useful interpretability to the optimization behavior.

**Weaknesses:**

W1: Selecting the best GPT-4 run for collaboration (p. 20) introduces potential selection bias, giving the fused model an unfair advantage and likely inflating reported performance. A more rigorous evaluation would use fixed or randomly sampled LLM outputs, or report mean and variance across multiple runs to ensure fairness.

W2: The assumption that the LLM’s score distribution follows a half-Gaussian form, and aligning the TSADM outputs to this distribution appears brittle and under-justified. The paper does not specify the parameterization of the mapping function M(⋅) or enforce monotonicity constraints. As a result, the alignment objective in Eq. (3) risks overfitting to an arbitrary density rather than learning a semantically meaningful correspondence.

W3: The paper omits precise mathematical definitions for 𝐷_intra and 𝐷_inter, referencing them only through qualitative visualizations in Fig. 3. Without explicit formulas or explanations of their metrics, scaling across variates, or windowing behavior, it is difficult to reproduce or interpret the reported sensitivity analyses.

**Questions:**

Q1: What is the functional class of M(⋅)? Is it constrained to be monotone? How do you avoid degenerate mappings (e.g., many inputs mapped to a single bin)? Please provide the exact architecture (layers/activations) and regularizers.

Q2: How many LLM runs were attempted per subset before picking the “best”? What were the decoding parameters?

Q3: In Table 3 (unseen distributions), did you tune hyperparameters on held‑out subsets only? Please clarify the selection protocol to ensure no test information was used.

---

> ### Author Response · Authors · 2025-11-20
> **Thank you for kind comments and Response 1**
>
> Thank you for your careful, thoughtful and helpful suggestions, which help us to improve the rigor of our experiments. The edited part according to your guidance is highlighted in green.
>
> **W1&Q2: choosing the best LLM run introduces choice bias and settings for LLM**
>
> Thank you for your kind reminder. We ran 2 times of LLM to choose the best in the original experiment. To improve the rigor of our experiments, we run LLM for 5 times totally and computes the mean and standard deviation of CoLLaTe's performance. The results are shown bellow, where CoLLaTe' s average performance also achieve the best F1 score across datasets.
>
> |               |           | Mustang   |           |           | Mackey    |           |           | Flight1   |           |           | Flight2   |           |
> | ------------- | --------- | --------- | --------- | --------- | --------- | --------- | --------- | --------- | --------- | --------- | --------- | --------- |
> |               | Pre       | Rec       | F1        | Pre       | Rec       | F1        | Pre       | Rec       | F1        | Pre       | Rec       | F1        |
> | Best Baseline | **1.000** | 0.891     | 0.935     | 0.829     | 0.960     | 0.882     | 0.690     | 0.947     | 0.768     | 0.749     | 0.742     | 0.746     |
> | CoLLaTe-mean  | 0.934     | **0.989** | **0.956** | **0.941** | **0.990** | **0.964** | **0.806** | **0.970** | **0.872** | **0.841** | **0.930** | **0.871** |
> | CoLLaTe-std   | 0.044     | 0.025     | 0.015     | 0.023     | 0.013     | 0.017     | 0.025     | 0.019     | 0.010     | 0.068     | 0.040     | 0.026     |
>
> The LLM decoding setting is
> ~~~
> client.chat.completions.create(
>     model="gpt-4o-mini",
>     messages=[Prompt],
>     temperature=1.0,
>     top_p=1.0,
>     n=1,
>     stream=False,
>     max_tokens=None,
>     presence_penalty=0.0,
>     frequency_penalty=0.0,
>     logit_bias=None,
>     user=None,
>     stop=None
> )

---

> ### Author Response · Authors · 2025-11-20
> **Thank you for kind comments and Response 2**
>
> **W2&Q1: unjustified Half-Gaussian assumption and Mapping function monotonicity**
>
> Thank you for your kind reminder. We agree that Half-Gaussian may not be suitable for all scenarios. The Half-Gaussian distribution can be replaced by any distribution with closed-form and derivable PDF. Our theory development does not depend on specific assumption of the distribution. Thus, we change the expression of $f(S)$ in our manuscript, which does not constrain it ot Half-Gaussian Assumption, as follows:
>
> *"We use a smooth and derivable function $f(S)$ to fit the LLM's anomaly score distribution."*
>
> Moreover, we supplement more experiments to show the goodness-of-fits of different distribution assumptions and show the CoLLaTe's performance under different reasonable distribution assumptions.
>
> *Effectiveness for half-gaussian assumption.* We add goodness-of-fits test of different anomaly score distribution assumption to verify that Half-Gaussian is a approvable assumption in the following table.
>
> |                        | D      | Bootstrap-P-Value |
> | ---------------------- | ------ | ----------------- |
> | Beta                   | 0.2023 | 0.5033            |
> | Logit-normal           | 0.2519 | 0.5633            |
> | Half-Gaussian          | 0.2428 | 0.4867            |
> | Gaussian Mixture Model | 0.1218 | 0.4967            |
>
> *Robustness for reasonable assumptions.* Moreover, we supplement more experiments to verify the robustness of CoLLaTe for different reasonable anomaly score distribution assumption in the following table.
>
> |               |           | Mustang   |           |           | Mackey    |           |           | Flight1   |           |           | Flight2   |           |
> | ------------- | --------- | --------- | --------- | --------- | --------- | --------- | --------- | --------- | --------- | --------- | --------- | --------- |
> |               | Pre       | Rec       | F1        | Pre       | Rec       | F1        | Pre       | Rec       | F1        | Pre       | Rec       | F1        |
> | Beta          | 0.933     | 0.992     | 0.961     | 0.851     | 0.992     | 0.915     | 0.792     | 0.972     | 0.860     | **0.965** | 0.848     | 0.885     |
> | Logit-normal  | 0.875     | **1.000** | 0.929     | 0.926     | **0.996** | 0.956     | **0.987** | 0.779     | 0.847     | 0.780     | **1.000** | 0.876     |
> | Half-Gaussian | **1.000** | 0.943     | 0.967     | **0.970** | **0.996** | **0.982** | 0.790     | **0.978** | 0.866     | 0.897     | 0.891     | 0.883     |
> | GMM           | 0.980     | **1.000** | **0.989** | 0.962     | **0.996** | 0.978     | 0.805     | 0.978     | **0.872** | 0.835     | 0.978     | **0.889** |
>
> *Architecture of $\mathcal{M}$ function and its Monotonicity.* The Mapping function is designed as the following Equation shows, where $R$ is time series representation obtained from a fixed TSADM. $\mathcal{M}(s_i)$ is a linear mapping function combined with a sigmoid function. Because both of them are monotonous and the time series representation part $R$ is a fixed constant (we do not change the learnable parameters of the pre-trained TSADM in our model), the $\mathcal{M}(s_i)$ is monotonous with respect to $s_i$.
>
> $$\bar{s}_i=\operatorname{Concat}(R,s_i), \mathcal{M}(s_i)=\operatorname{Sigmoid}(W\bar{s}_i)$$
>
> *How to avoid degenerate mappings?* We avoid the degenerate mapping by the regularization item in alignment loss function (Eq.(3)), where we add the item in the following Equation (we split it into two lines to compile it correctly in openReview, thank you for your understanding). This item force the divergence of aligned scores (scores after mapping) close to the divergence of LLM anomaly score distribution. Thus, the mapped scores will not fall in one bin.
>
> $$\hat{\lambda}_2 \cdot$$
>
> $$(\frac{1}{n-1}\sum_{i=1}^n(\mathcal{M}(s_i) - \mu_M)^2 - \hat\sigma^2)^2$$
>
> **W3: formal definition of $D_{intra}$ and $D_{inter}$**
>
> Thank you for your kind suggestion. We add formal definition of $D_{intra}$ and $D_{inter}$ as follows in the main text of resubmitted paper.
>
> $$D_{intra}(i)=\frac{1}{t-1}\sum_{j\in [0,t]\backslash i}d(P[i],P[j])$$
>
> $$D_{inter}(i)=\frac{1}{T/t-1}\sum_{j\in[0,T/t]\backslash i}d(P[k][i],P[k][j])$$
>
> **Q3: unseen distribution experiment settings**
>
> The unseen distribution experiment involves a source domain and a target domain. Both training set and validate set are from the source domain, while the test set is from the target domain. Specifically, the training set and validate set comes from a same dataset, where we divide the first 5/6 as the training set and the remaining 1/6 as the validate set. The test set is from another dataset (e.g. monitoring data from other plane, other cluster, etc.) Thus, it can be ensured that the model is not exposed to the distribution of test set during the training process.

---

> > ### Comment · Reviewer_ct8Q · 2025-11-25
> >
> > Thank you for your response. I appreciate the authors’ efforts to address my concerns, and I have accordingly raised my score.

---

> > > ### Author Response · Authors · 2025-11-26
> > >
> > > Thank you for your good feedback. We are glad to polish our paper according to your constructive guidance.

---

### Official Review · Reviewer_2vmL · 2025-10-30

**Soundness:** 3
**Presentation:** 3
**Contribution:** 2
**Rating:** 6
**Confidence:** 4

**Summary:**

The paper introduces CoLLaTe, a collaborative framework that integrates Large Language Models (LLMs) with task-specific anomaly detection models (TSADMs) to improve time-series anomaly detection.

**Strengths:**

1. The paper explicitly identifies two bottlenecks—domain misalignment and error accumulation—and addresses them through clearly designed modules backed by theoretical proofs.
2. CoLLaTe achieves state-of-the-art F1 scores on multiple datasets and shows robustness to distribution shifts, which strengthens the practical relevance of the framework.
3. The theoretical results (Theorem 1, Lemma 1, Theorem 2) are well-connected to experimental observations, lending credibility to the proposed loss design.

**Weaknesses:**

1. While the idea of combining LLMs and specialized models is timely, the actual implementation—alignment + weighted fusion—is relatively straightforward.
2. LLM inference is computationally heavy; integrating it with TSADM could raise latency issues.
3. The proofs rely on smoothness and distributional assumptions that may not hold in practice (e.g., half-Gaussian score distribution).

**Questions:**

1. Are the LLM outputs frozen or fine-tuned jointly with the TSADM and conditional network, and why?
2. How sensitive is the alignment module to the choice of the fitted distribution (half-Gaussian)? Would a non-parametric mapping (e.g., histogram matching) perform similarly?
3. Can the aligned scores or the conditional network outputs be interpreted to explain why a specific time point is deemed anomalous?

---

> ### Author Response · Authors · 2025-11-20
> **Thank you for good feedbacks and Response 1**
>
> We are very grateful for your good, thoughtful and helpful comments. We are pleased to polish our paper according to your guidance. The edited part in the resubmitted manuscript according to your suggestion is highlighted in brown.
>
> **W1: straightforward idea**
>
> We appreciate this observation. Indeed, we intentionally designed a conceptually simple pipeline, because simplicity in the high-level structure does not imply simplicity in achieving reliable performance.
>  The difficulty lies not in proposing a alignment + fusion pipeline, but in designing a principled and effective alignment mechanism, handling distribution mismatch, and avoid error accumulation observed in fusion process. Our method provides concrete and technically non-trivial solutions to these challenges. Thank you for your understanding.
>
> **W2: heavy overhead of LLM**
>
> Thank you for your thoughtful comment. We agree that LLMs introduce additional latency, and therefore CoLLaTe is not intended for scenarios that require strict real-time response. Instead, CoLLaTe is better suited for situations where accuracy takes priority over speed and where substantial domain expertise is required to identify anomalies.
>
> For example, during ground tests after aircraft assembly, engineers are willing to spend more time to ensure the aircraft’s condition is assessed correctly, because missing an anomaly may lead to severe consequences during subsequent flight tests, such as plane crash and casualty. Similar cases include safety checks before restarting large industrial equipment, inspections of power systems before they go online, or pre-mission diagnostics for UAVs and ships. In these scenarios, taking additional time is acceptable—but making an incorrect decision is not.
>
> **W3: the proofs rely on smoothness and distribution assumption**
>
> *Smoothness Assumption.* Thank you for your kind comment. We recognize this assumption may not be practical and we rewrite the proof that depended on this Assumption in Appendix.A.3 (Verify Assumption 5.2 from a priori work). Now, it does not depend on the smoothness assumption and we remove this assumption from the main text.
>
> *Distribution Assumption.* The half-Gaussian distribution can be replaced by any distribution that admits a closed-form and derivable PDF. Our theory development does not depend on specific formation of $f(S)$. Thus, we change the expression of $f(S)$ in our manuscript, which does not constrain it to half-gaussian assumption as follows:
>
> *"We use a derivable function $f(S)$, which has a close format, to fit the LLM's anomaly score distribution"*
>
> Moreover, we conducted goodness-of-fit tests under different distributional assumptions to verify that that Half-Gaussian distribution is approvable and supplement experiments of CoLLaTe on different distribution assumptions.
>
> |                        | D      | Bootstrap-P-Value |
> | ---------------------- | ------ | ----------------- |
> | Beta                   | 0.2023 | 0.5033            |
> | Logit-normal           | 0.2519 | 0.5633            |
> | Half-Gaussian          | 0.2428 | 0.4867            |
> | Gaussian Mixture Model | 0.1218 | 0.4967            |
>
> We recognize Gaussian Mixture Model is a better alternative. Thus, we conduct experiments on different distribution assumptions to verify: 1) our model is robust to different reasonable distribution assumptions and we can pick the most suitable one under different scenarios; 2) replacing Half-Gaussian with GMM can obtain better performance.
>
> |               |           | Mustang   |           |           | Mackey    |           |           | Flight1   |           |           | Flight2   |           |
> | ------------- | --------- | --------- | --------- | --------- | --------- | --------- | --------- | --------- | --------- | --------- | --------- | --------- |
> |               | Pre       | Rec       | F1        | Pre       | Rec       | F1        | Pre       | Rec       | F1        | Pre       | Rec       | F1        |
> | Beta          | 0.933     | 0.992     | 0.961     | 0.851     | 0.992     | 0.915     | 0.792     | 0.972     | 0.860     | **0.965** | 0.848     | 0.885     |
> | Logit-normal  | 0.875     | **1.000** | 0.929     | 0.926     | **0.996** | 0.956     | **0.987** | 0.779     | 0.847     | 0.780     | **1.000** | 0.876     |
> | Half-Gaussian | **1.000** | 0.943     | 0.967     | **0.970** | **0.996** | **0.982** | 0.790     | **0.978** | 0.866     | 0.897     | 0.891     | 0.883     |
> | GMM           | 0.980     | **1.000** | **0.989** | 0.962     | **0.996** | 0.978     | 0.805     | 0.978     | **0.872** | 0.835     | 0.978     | **0.889** |

---

> ### Author Response · Authors · 2025-11-20
> **Thank you for good feedbacks and Response 2**
>
> **Q1: are the LLM frozen or fine-tuned with TSADM?**
>
> The LLM output is frozen intentionally because following reasons. Intuitively, the role of LLM is to provide high-level expert knowledge, for different tasks we just need to change the expert knowledge in prompt and do not need to fine-tune LLM with TSADM on the training set. Furthermore, allowing the LLM to update its huge amount of parameters would risk overfitting to the limited anomaly-detection training set.
>
> Practically, fixed LLM parameters has two benefits:
>  (1) Stability and convergence: Freezing LLM output avoids stochastic shift in LLM anomaly score distribution, ensuring alignment is solely driven by the lightweight alignment module, which can be converged more stably.
>  (2) Efficiency: Fine-tuning billions-parameter models can introduce significant GPU overhead, and optimization instability. Our design keeps the training process of CoLLaTe lightweight.
>
> Empirically, we our experiments proved that CoLLaTe with frozen-LLM already achieves strong performance. This supports our assumption that the LLM’s pretrained knowledge and expert knowledge in prompt is sufficient.
>
> **Q2: alignment module sensitivity to distribution assumption**
>
> As discussed in **W2**, the alignment module is not sensitive to the Half-Gaussian distribution assumption, other distributions can also work well. However, non-parametric mapping can not be used in alignment module, because in alignment module we need a closed-form PDF to train it.
>
> **Q3: the interpretability of aligned scores and the conditional network**
>
> The aligned scores and conditional network outputs can not be directly interpreted to explain why a time point is deemed as anomaly. But we can obtain an interpretation of CoLLaTe by the following step:
>
> 1. Compute $D_{inter}$ and $D_{intra}$ and see which model contributes more to the final result.
>
> 2. If the TSADM contributes more to the final result: we can compute reconstruction error at each dimension and localize the root cause of the anomaly.
>
> 3. If LLM contributes more to the final result: we can get interpretation from LLM's response by removing the requirement "only output the anomaly score". Then, we can get the chain of thought of LLM.
>
>
>
> We provide an example of LLM's response in the following:
>
>  *To determine if there are many tasks slow down at time slot 0 compared to the other time slots in the given sliding window, we can analyze the ratios of tasks for each time slot.*
>
> *Here's the breakdown for time slot 0 based on the given input data:*
>
> *- Time slot 0: [0.75, 0.08299999684095383, 0.16699999570846558, ...]*
>
> *- The first value (0.75) indicates that 75% of tasks have a duration in the range [0, 5).*
>
> *Now, we can compare this value to the other time slots to assess the overall distribution of tasks:*
>
> *From the first five time slots (for example):*
>
> *- Time slot 1: [0.4000000059604645, ...]*
>
> *- Time slot 2: [0.5, ...]*
>
> *- Time slot 3: [1.0, ...]*
>
> *- Time slot 4: [0.0, ...]*
>
> *In observing the first dimension, a value of 0.75 in time slot 0 is relatively high compared to slots 1, 2, and 4. However, time slot 3 has a value of 1.0, which indicates all tasks are in the [0, 5) duration.*
>
> *One approach to calculate a probability score for slot 0 is to evaluate the distribution of tasks across all time slots considering an anomaly detection score.*
>
> *Given there are examples with specific scores (0.000000 for time slot 0 indicating no slowdown) and 1.000000 for time slot 3 indicating a significant slowdown, we can infer that:*
>
> *- A score approaching 0 indicates normal conditions.*
>
> *- A score significantly higher than 0 indicates detected anomalies or slowdowns.*
>
> *Considering the distribution:*
>
> *- Time slot 0 has a rather high concentration of tasks with fast durations, while other time slots show varied distributions.*
>
> *Thus, the probability that time slot 0 has many tasks slow down compared to others can be assigned a score consistent with lower than the typical anomaly score.*
>
> *Based on the comparative analysis, I would assign a score of approximately 0.1 for time slot 0's probability of having many tasks slowed down compared to the overall data:*
>
> *Final output:*
>
> *Anomaly Score: 0.1*

---

### Official Review · Reviewer_NniG · 2025-11-01

**Soundness:** 2
**Presentation:** 2
**Contribution:** 2
**Rating:** 4
**Confidence:** 4

**Summary:**

The paper proposes **CoLLaTe**, a framework that collaborates a task-specific time-series anomaly detector (TSADM) with an LLM. Two main components are introduced:
1. a **score-alignment module** to map TSADM/LLM anomaly scores into a unified semantic/distributional space;
2. a **collaborative loss** intended to mitigate “error accumulation” when combining the two sources.
    Experiments on four datasets (two public, two private) show that naïve combination can hurt performance, while alignment + collaborative loss recover and sometimes improve results (small gains on Mustang, larger gains on Mackey/Flight). The motivation is practical and the ablations are informative, but related work coverage, key baselines, theoretical rigor, and cost–benefit analysis are insufficient.

**Strengths:**

- **Well-motivated engineering problem:** addresses the real friction of score semantic mismatch and error accumulation when mixing TSADM and LLM signals.
- **Clear components & useful ablations:** alignment + collaborative loss are shown necessary (naïve fusion degrades; adding components repairs performance).
- **Some empirical gains:** meaningful improvements on several datasets; the workflow matches practical “alarm triage / false-positive reduction” use cases.
- **Potential for deployment:** conceptually compatible with industrial pipelines.

**Weaknesses:**

1. **Theory: unclear definitions and shaky derivations**


- **Alignment objective.** Main text moves from a binned, non-differentiable objective to a smooth surrogate but skips formalities. Let (f) be the target density (paper uses half-Gaussian). The non-diff objective is essentially
$$
    \min_{M}; -\sum_{i=1}^N \tfrac{c_i}{\sum_j c_j}\log!\Big(\int_{(i-1)/N}^{i/N} f(S),dS\Big)
    ;+; \lambda_1!\Big(\tfrac{1}{n}\sum\nolimits_{k} M(s_k)-\hat\mu\Big)^2
    ;+; \lambda_2(\cdots)^2,
$$
    where (c_i) counts mapped scores falling in bin (i). The paper then states (for (N!\to!\infty)) the surrogate
$$
    \min_{M}; -\tfrac{1}{n}\sum\nolimits_{k}\log f(M(s_k)) + \lambda_1(\cdots)^2+\lambda_2(\cdots)^2.
$$
    However, the derivation omits the constant (\log(1/N)) term and provides no error bound (e.g., Riemann-sum or dominated-convergence justification) for replacing the binned cross-entropy with the continuous NLL. Please provide a formal proposition with assumptions and a uniform convergence (or at least consistency) statement.

- **Choice of (f).** Using a half-Gaussian for scores (S $\ge$ 0) is plausible but not justified. Given scores lie on a bounded interval in practice, Beta / logit-normal or isotonic calibration might be more appropriate. Please provide goodness-of-fit tests (e.g., KS, AD) and a robustness study across families; otherwise alignment risk is model-misspecified.

- **Collaborative loss / “error accumulation” theorem.** The paper states the MSE-based combination yields an optimum $\hat S^\star = y + \lambda_1\varepsilon_s + \lambda_2\varepsilon_S$ and then lower-bounds $\mathbb{E}[(\hat S^\star - y)^2]$ by $(\lambda_1\mu_s+\lambda_2\mu_S)^2$ (Jensen). This ignores variance and covariance:
    $$
    \mathbb{E}[(\hat S^\star-y)^2]
    = \lambda_1^2(\sigma_s^2+\mu_s^2)+\lambda_2^2(\sigma_S^2+\mu_S^2)+2\lambda_1\lambda_2\operatorname{Cov}(\varepsilon_s,\varepsilon_S),
    $$
    which reduces to $(\lambda_1\mu_s+\lambda_2\mu_S)^2$ only under **zero variance** (or degenerate) conditions. Moreover the stationarity condition mixes $\hat S$ as a deterministic function of parameters with random errors $\varepsilon$. Please **restate the probability space**, clarify whether $\hat S$ depends on sample noise at optimum, and **include variance/covariance** in the bound (or give conditions—unbiasedness, independence, bounded variance—under which your inequality holds).

- **Undefined key quantities.** (D_{\text{intra}}) / (D_{\text{inter}}) are described verbally (“within-patch / across-patch distances”) but **lack formulas**. For example, if (x_t) are embeddings and (p(t)) is the patch index,
    $$
    D_{\text{intra}}(t)=\tfrac{1}{|P(t)|-1}!\sum_{j\in P(t)\setminus{t}}! d(x_t,x_j),
    \quad
    D_{\text{inter}}(t)=\tfrac{1}{K-1}!\sum_{k\ne p(t)}! d(\mu_{p(t)},\mu_k),
    $$
    with (d) (Euclidean/DTW/cosine) and (\mu_\cdot) specified. Please provide exact definitions, normalization, and metric choice in the main text.

- **Convergence claim.** The $(O(T^{-1/4}))$ rate borrowed from prior work requires Lipschitz smoothness, bounded stochastic gradients/noise, step-size schedule, etc. The paper should verify each assumption for the specific loss (including the alignment regularizers) and report constants (at least qualitatively).

2. **Positioning & baselines**
- Missing **closest paradigms**: (i) _LLM→TSAD distillation_ (training-time guidance; different collaboration locus), (ii) _TSAD→LLM two-stage refinement_ (post-hoc vetting). Without these, contribution boundaries blur.
- Missing **simple fusions**: score avg / max / vote and a learned meta-fuser (e.g., logistic regression/XGBoost on ($S_{\text{TSADM}},S_{\text{LLM}}$) with calibration) are standard sanity checks to justify your more complex design.

2. **Evaluation & practicality**
- **Cost–benefit** is unclear: when strong TSADM already yields high F1, LLM adds only +2–3 points in places, yet latency/API cost appears substantial. Provide throughput/latency/cost vs. gain curves and scenarios where LLM is decisively worth it.
- Some narrative claims (e.g., “LLM excels at point anomalies”) are not consistently supported; include a breakdown by anomaly type/length.

**Questions:**

1. **Add closest & simple baselines:** Include _LLM→TSAD distillation_, _TSAD→LLM two-stage refinement_, and simple score fusion (avg / max / vote). Report gaps relative to CoLLaTe.
2. **Theory fixes:**
    - Give formal definitions and computation for $(D_{\text{intra}})$ and $(D_{\text{inter}})$.
    - Re-state/prove the error-accumulation result with a consistent probability space and explicit variance terms.
    - Justify the alignment distribution with goodness-of-fit tests and robustness analysis.
3. **Where does LLM truly help?** Provide case studies and breakdowns (point vs. contextual, short vs. long anomalies) showing indispensable improvements.
4. **Explain weak TSFM baselines:** Document implementations, tuning budgets, and why they underperform here.

---

> ### Author Response · Authors · 2025-11-20
> **Thank you for valuable comments and Response 1**
>
> We are very thankful that you not only give comments but also give suggesions on how to modify our manuscript to improve its quality in your review. We are pleased to have the opportunity to revise the manuscript based on your kind suggestions and to make the proof more rigorous. The edited parts in the manuscript according to your suggestion are highlighted in blue.
>
> **W1.1: Riemann-sum justification and uniform convergence statement of Alignment objective**
>
> Thank you for your kind suggestion. We polish the proof in appendix A.1 in our resubmitted manuscript and propose a new proposition, where we prove when $N$ approaches infinite, $-\frac{1}{n}\sum_{i=1}^{n}[\log(f(\mathcal{M}(s_i)))-\log{N}]$ can approximate $-\sum_{i=1}^N  \frac{c_i}{\sum_{j=1}^N c_j} \log \int_{(i-1)b/N}^{ib/N} f(S)dS$, with $o(1)$ upper error bound.
>
> And we can transform the original non-differential objective function into following, where $r_1$ and $r_2$ are the regularization items of mean and variance (it is too complex for openReview to compile it correctly, please refer to the resubmitted manuscript for details, thank you for your understanding)
>
> $$
> \min_{\mathcal{M}} \; -\frac{1}{n}\sum_{i=1}^{n}[\log(f(\mathcal{M}(s_i)))-\log N]+r_1+r_2
> $$
>
>
> For any $\epsilon>0$, we can find an $N_0$, if $N>N_0$, the error between the smooth surrogate loss function and the original non-differential one is below $\epsilon$. Thus, when optimizing $\mathcal{M}$, we can set $N$ as a big enough constant, which allows the error below the acceptable error limit. Then, the $-\log{N}$ (the $\log{\frac{1}{N}}$ item) is a constant and will not impact the minimization process and the optimal solution of the objective function. Thus, we omit it and obtain the smooth objective function in the original manuscript.
>
> *Convergence guarantee.* There are some misleading expressions in the original manuscript, and we clarify them as follows. After the simplication process above, the summation in our objective contains $n$ terms, where $n$ denotes the number of data samples, rather than the number of bins $N$. That is because for most of bins, $c_i=0$ and can be omitted in our simplification process above. In the summation, each term is a finite scalar, so the objective is simply a finite sum of finite quantities. Therefore, the optimization objective is well-defined and guaranteed to be convergent.
>
> **W1.2&Q2.3: Justification of $f$ choice**
>
> Thank you for your suggestions, we provide goodness-of-fits test (KS) across half-Gaussian/Beta/logit-normal/isotonic calibration/ Gaussian Mixture Model (GMM, number of components is 5). According to the goodness-of-fits test, GMM can fit the distribution best. Since our theory development actually does not rely on specific formation of $f(S)$ but only requires it is derivable with close format, we change its expression in our manuscript as follows:
>
> *"We use a derivable function $f(S)$, which has a close format, to fit the LLM's anomaly score distribution"*
>
> |                           | D      | Bootstrap-P-Value |
> | ------------------------- | ------ | ----------------- |
> | Beta                      | 0.2023 | 0.5033            |
> | Logit-normal              | 0.2519 | 0.5633            |
> | Isotonic                  | 0.1225 | 0.4933            |
> | Half-Gaussian             | 0.2428 | 0.4867            |
> | Mixture of Gaussian Model | 0.1218 | 0.4967            |
>
> (Robustness verification) We rerun our model with Beta distribution, Logit-normal and GMM, and list their performance bellow. (Because the CDF of Isotonic calibration is a step function and will encounter problem when differentiating, we do not include it here). Since GMM can fit the LLM anomaly score distribution better, it can achieve better performance in most of datasets. Besides, our model is also robust for other reasonable assumptions.
>
> |               |           | Mustang   |           |           | Mackey    |           |           | Flight1   |           |           | Flight2   |           |
> | ------------- | --------- | --------- | --------- | --------- | --------- | --------- | --------- | --------- | --------- | --------- | --------- | --------- |
> |               | Pre       | Rec       | F1        | Pre       | Rec       | F1        | Pre       | Rec       | F1        | Pre       | Rec       | F1        |
> | Beta          | 0.933     | 0.992     | 0.961     | 0.851     | 0.992     | 0.915     | 0.792     | 0.972     | 0.860     | **0.965** | 0.848     | 0.885     |
> | Logit-normal  | 0.875     | **1.000** | 0.929     | 0.926     | **0.996** | 0.956     | **0.987** | 0.779     | 0.847     | 0.780     | **1.000** | 0.876     |
> | Half-Gaussian | **1.000** | 0.943     | 0.967     | **0.970** | **0.996** | **0.982** | 0.790     | **0.978** | 0.866     | 0.897     | 0.891     | 0.883     |
> | GMM           | 0.980     | **1.000** | **0.989** | 0.962     | **0.996** | 0.978     | 0.805     | 0.978     | **0.872** | 0.835     | 0.978     | **0.889** |

---

> ### Author Response · Authors · 2025-11-20
> **Thank you for valuable comments and Response 2**
>
> **W1.3&Q2.2: Error accumulation theorem**
>
> 1. *The variance item*: Thank you for your kind reminder. There are some ambiguous statements in the manuscript that causes misunderstanding and we fix it now. Here is a correct proof:
>
> $$\hat{S}^*(\mathcal{P},\epsilon_s,\epsilon_S)=y+\lambda_1\epsilon_s+\lambda_2\epsilon_S$$
>
> Thus, we have:
>
> $$\mathbb{E}[(\hat{S}^*(\mathcal{P},\epsilon_s,\epsilon_S)-y)^2]=\mathbb{E}[(\lambda_1\epsilon_s+\lambda_2\epsilon_S)^2]$$
>
> Since $Var(X)=\mathbb{E}[(X-\mathbb{E}(X))^2]=\mathbb{E}[X^2]-(\mathbb{E}[X])^2\ge 0$,  we have $\mathbb{E}[X^2]\ge (\mathbb{E}[X])^2$.
>
> Thus:
>
> $$\mathbb{E}[(\lambda_1\epsilon_s+\lambda_2\epsilon_S)^2] \geq \left|\mathbb{E}[\lambda_1\epsilon_s+\lambda_2\epsilon_S]\right|^2$$
>
> Since $\epsilon_1 \sim \mathcal{D}_s(\mu_s,\sigma_s), \epsilon_2 \sim \mathcal{D}_S(\mu_S,\sigma_S)$, we have $\mathbb{E}[\lambda_1\epsilon_s+\lambda_2\epsilon_S]=\lambda_1\mu_s+\lambda_2\mu_S$
>
> Thus, we have:
>
> $$\left| \mathbb{E}[\lambda_1\epsilon_s+\lambda_2\epsilon_S] \right|^2=(\lambda_1\mu_s+\lambda_2\mu_S)^2$$
>
>  (note: here is a square of the expectation rather than an expectation of a square item, thus we do not introduce covariance item)
>
> Thus, $\mathbb{E}[(\hat{S}^*(\mathcal{P},\epsilon_s,\epsilon_S)-y)^2] \geq (\lambda_1\mu_s+\lambda_2\mu_S)^2$
>
> 2. *Restate the probability space*
>
> Thank you for your kind reminder. We restate the probability space by changing $\hat{S}(\mathcal{P})$ to $\hat{S}(\mathcal{P},\epsilon_s,\epsilon_S)$ and  prove that introducing random variables into $\hat{S}(\mathcal{P})$ does not affect the conclusion of accumulated error. We put the detailed proof in Appendix. A.2, where we can obtain the following equation according to KKT condition.
>
> $$2\lambda_1(y+\epsilon_s-\hat{S}^*(\mathcal{P},\epsilon_s,\epsilon_S))$$
>
> $$+2\lambda_2(y+\epsilon_S-\hat{S}^*(\mathcal{P},\epsilon_s,\epsilon_S))=0$$
>
> Reorganized the above equation, we can obtain:
>
> $$\hat{S}^*(\mathcal{P},\epsilon_s,\epsilon_S)=y+\lambda_1\epsilon_s+\lambda_2\epsilon_S$$
>
> Follow, the proof in 1, we can prove that introducint random variables does not affect the conclusion of accumulated error, where $\mathbb{E}[(\hat{S}^*(\mathcal{P},\epsilon_s,\epsilon_S)-y)^2] \geq (\lambda_1\mu_s+\lambda_2\mu_S)^2$.
>
> **W1.4&Q2.1: Definition of key quantities**
>
> We add the formal definition of $D_{inter}$ and $D_{intra}$ as follows:
>
> We split a time series $x\in R^{T\times d}$ into patches $P\in R^{t\times T/t \times d}$. Then, we define the $D_{inter}$ and $D_{intra}$ as follows, where $d(\cdot , \cdot)$ is defined as the Euclidean distance and $k$ is the index of the patch to which the $i$-th time step belongs.
>
> $$D_{intra}(i)=\frac{1}{t-1}\sum_{j\in [0,t]\backslash i}d(P[i],P[j])$$
>
> $$D_{inter}(i)=\frac{1}{T/t-1}\sum_{j\in[0,T/t]\backslash i}d(P[k][i],P[k][j])$$
>
> **W1.5: Convergence claim**
>
> Thank you for your kind suggestion. We reformat the proof in Appendix. A.3 and verify the collaborative loss functions satisfies every Assumptions in [1]. As for the alignment loss function, which is used to align the distribution of TSADM anomaly score with the ones of LLM, because the TSADM and LLM are fixed in our model's training process (i.e. the training process do not affect the anomaly score distributions from these models), the optimal mapping function to minimize the alignment objective is irrelevant to the training process of conditional neural network. Thus, the alignment loss function and collaborative loss function can be used separately. And in Lamma 3, we prove the collaborative loss function can converge with $O(T^{-1/4})$. Thank you for your understanding.
>
> [1] Bu, Zhiqi, et al. "Automatic clipping: Differentially private deep learning made easier and stronger." *Advances in Neural Information Processing Systems* 36 (2023): 41727-41764.

---

> ### Author Response · Authors · 2025-11-20
> **Thank you for valuable comments and Response 3**
>
> **W2&Q1: closest and simple baselines**
>
> We list the performance of TSADM -> LLM, LLM -> TSADM and simple score fusion strategies bellow, where CoLLaTe still achieve the best F1 score.
>
> - LLM -> TSADM: We first train a TSADM on a given dataset. Then, we fine-tune the TSADM with the anomaly scores of an LLM by distillation learning.
> - TSADM -> LLM: We integrate the anomaly score output by a TSADM with the original LLM prompt.
> - Max/Avg: We compute the Max/Avg of the anomaly scores from a TSADM and a LLM.
> - Vote: We use a TSADM and two randomly sampled LLM output to vote.
>
> |              |           | Mustang   |           |           | Mackey    |           |           | Flight1   |           |           | Flight2   |           |
> | ------------ | --------- | --------- | --------- | --------- | --------- | --------- | --------- | --------- | --------- | --------- | --------- | --------- |
> |              | Pre       | Rec       | F1        | Pre       | Rec       | F1        | Pre       | Rec       | F1        | Pre       | Rec       | F1        |
> | TSADM -> LLM | 0.696     | 0.889     | 0.571     | 0.259     | **1.000** | 0.411     | 0.375     | 0.857     | 0.522     | 0.416     | 0.877     | 0.458     |
> | LLM -> TSADM | 0.900     | 0.927     | 0.889     | 0.918     | 0.787     | 0.820     | 0.793     | 0.819     | 0.706     | 0.667     | 0.923     | 0.774     |
> | Max          | 0.857     | 0.861     | 0.845     | 0.768     | **1.000** | 0.863     | 0.845     | 0.855     | 0.794     | 0.778     | 0.824     | 0.799     |
> | Avg          | 0.874     | 0.907     | 0.881     | 0.812     | **1.000** | 0.891     | **0.898** | 0.734     | 0.776     | 0.563     | **1.000** | 0.720     |
> | Vote         | 0.076     | **1.000** | 0.141     | 0.177     | **1.000** | 0.301     | 0.157     | 0.889     | 0.267     | 0.112     | 0.876     | 0.199     |
> | CoLLaTe      | **1.000** | 0.943     | **0.967** | **0.970** | 0.996     | **0.982** | 0.790     | **0.978** | **0.866** | **0.897** | 0.891     | **0.883** |
>
> **W3.1：Cost benefit**
>
> Thank you for your kind suggestion.
>
> *Cost-gain curve*. Since the LLM time overhead mainly comes from Internet latency and force sleep command of the LLM API, which is uncontrollable and irrelevant to the performance of LLM, it is hard to draw a curve of time overhead vs. gain curves. Instead, we compare the time overhead of CoLLaTe and CoLLaTe-w/o LLM, as well as their F1 score in the following table. Since the Mackey dataset is based on the Mackey-glass equation and the Mustang dataset is to detect cluster-wide slow down from task execution time distribution, detecting anomalies from them depends more on the data fluctuation patterns rather than expert knowledge. As for Flight 1 and Flight 2, since they are monitoring data collected from aircraft, detecting anomaly from them depends on prior knowledge about aircraft design. Thus, incorporating LLM can improve F1 score on Flight 1 and Flight 2.
>
> |                                      | Mustang | Mackey | Flight1 | Flight2 |
> | ------------------------------------ | ------- | ------ | ------- | ------- |
> | $T_{CoLLaTe}-T_{w/o-LLM}$ (s/sample) | 1.31    | 1.24   | 1.26    | 1.41    |
> | $F1_{CoLLaTe}-F1_{w/o-LLM}$ (%)      | 2.9     | 2.5    | 17.2    | 18.1    |
>
> *Suitable Application Scenarios.* CoLLaTe is better suited for situations where accuracy matters more than strict real-time response and require lots of expert knowledge to detect anomaly. For example, during ground tests after aircraft assembly, engineers prefer spending extra time to ensure the aircraft's health is assessed correctly, because a missed anomaly could lead to serious accidents during aircraft flight tests, which can lead to a plane crash and people death. Similar situations include safety checks before restarting large industrial machines, inspections of power systems before they go online, or pre-mission checks for UAVs and ships. In all these cases, taking a little longer is acceptable, but making a wrong decision is not.

---

> ### Author Response · Authors · 2025-11-20
> **Thank you for your valuable comments and Reponse 4**
>
> **W3.2&Q3: where does LLM truly help?**
>
> We collect the F1 score of CoLLaTe and CoLLaTe-w/o-LLM on contextual/point, long/short anomalies and list them bellow, where we can see LLM show indispensable improvements on point anomaly and short anomalies. As for contextual anomaly and long anomalies, LLM contributes little. We attribute this to the "lost in the middle effect" of LLM [1]. Because when detecting contextual and long anomalies, the LLM need to compare the anomalies with longer time series contexts, the "lost in the middle effect" can hinder LLM successfully detect such anomalies.
>
> |                 | Point     | Context   | Short     | Long      |
> | --------------- | --------- | --------- | --------- | --------- |
> | CoLLaTe         | **0.891** | 0.341     | **0.872** | 0.981     |
> | CoLLaTe-w/o-LLM | 0.848     | **0.358** | 0.798     | **0.987** |
>
> Besides, we add case studies in Appendix. A.21 to analyze where LLM can truly help by an visualized example.
>
> [1] Liu, Nelson F., et al. "Lost in the middle: How language models use long contexts." *Transactions of the Association for Computational Linguistics* 12 (2024): 157-173.
>
> **Q4: Explain TSFM weak performance**
>
> *Document implementation details.*
>
> Dataset Division: We use the first 40% samples in each dataset as the training set, use the next 10% samples in each dataset as validation set and use the last 50% samples in each dataset as the test set.
>
> Model Download: We follow the official GitHub instructions of each method to obtain their pretrained models.
>
> Anomaly Detection: For OneFitsAll, we run the "run.py" recommended in the github to detect anomaly. As for Moment and Timer, we use the prediction error (MSE) as the anomaly score and use POT algorithm to find the threshold.
>
> *Explanation of the weak performance.*
>
> The weak performance of TSFM can come from the overgeneralization problem in anomaly detection. Those TSFM are used as prediction-based anomaly detection method in our experiment. Since they are trained on lots of dataset in pretraining process, it has good generalization across different data patterns, which is verified by the zero-shot prediction performance reported by many TSFM papers [1,2]. However, this strength in zero-shot prediction can be a shortage when using it as a prediction-based anomaly detection, because it can also fit the anomaly patterns well. In such situation, the prediction error (used as anomaly score) of anomalies can also be small and is hard to distinguish from normal samples. We draw the anomaly score distributions of anomaly and normal patterns for TSFM in Appendix. A.22, where their distributions are mostly overlapped and we can not distinguish them.
>
> [1] Goswami, Mononito, et al. "Moment: A family of open time-series foundation models." ICML 2024.
>
> [2] Liu, Yong, et al. "Timer: Generative pre-trained transformers are large time series models." ICML 2024.

---

### Author Response · Authors · 2025-11-30
**Close Remark and Sincere Gratitude to ACs' and PCs' Extra Burden Caused by the Accident**

Dear PCs, ACs, and Reviewers,

We sincerely thank all committees for their tremendous efforts devoted to this paper under such an unexpected accident. We truly appreciate the professionalism, patience, and extra workload undertaken by the ACs and PCs in organizing and safeguarding the review process. We are also deeply grateful for the constructive, careful, and high-quality comments provided by all reviewers.

To ease the burden on the AC, we objectively and concisely summarize the major common review concerns and our rebuttal progress before the information leakage incident. Notably, before the incident, Reviewer ct8Q had already responded positively to our rebuttal and **increased the score from 4 to 6**.

## **Strengths**

1. **Practical relevance and real-world motivation.**
    The paper tackles a well-motivated and practically important engineering problem (Reviewer NniG-S1), with strong practical value and real-world applicability (Reviewer ct8Q-S2, Reviewer-S4).
2. **Consistent experimental validation and reasonable loss design.**
    Extensive experiments on multiple real-world datasets, including ablations and sensitivity studies, consistently validate the effectiveness and superiority of the method (Reviewer ct8Q-S3, 2vmL-S2, NniG-S3).
    Moreover, experimental results align well with theoretical analysis, jointly supporting the rationality of the loss design (Reviewer 2vmL-S3, Reviewer ct8Q-S4).
3. **Identification of key bottlenecks in LLM–TSADM collaboration.**
    The paper identifies two critical bottlenecks in LLM–TSADM collaboration and addresses them effectively (Reviewer 2vmL-S1, ct8Q-S1).

## **Weaknesses and Responses**

1. **Assumption validity**

**1.1. Half-Gaussian assumption of LLM anomaly scores**
 (Reviewer ct8Q-W2, Reviewer NniG-W-1.3, Reviewer 2vmL-W3)

**Response:**
 We conducted KS-statistics goodness-of-fit tests, whose p-values support the validity of this assumption on our datasets. More importantly, our theoretical derivation does not rely on this specific distribution. The half-Gaussian can be replaced with any differentiable distribution with a closed form. Accordingly, we additionally tested Beta, Gaussian Mixture, and Logit-normal distributions, and compared both performance and KS-statistics.
 *This response received positive feedback from Reviewer ct8Q before the information leakage.*

**1.2. Smoothness assumption of the loss function**
 (Reviewer 2vmL-W3)

**Response:**
 We revised the proof to remove the smoothness requirement entirely (Appendix A.3).

2. **Latency introduced by LLMs**

(Reviewer 2vmL-W2, Reviewer NniG-W2)

**Response:**
 We agree that LLMs introduce extra latency. Therefore, CoLLaTe is not designed for strict real-time scenarios, but for settings where accuracy outweighs speed and strong domain knowledge is required.
 For instance, in post-assembly aircraft ground testing, engineers prefer higher diagnostic accuracy even at the cost of time, since missing anomalies may cause catastrophic safety risks in later flight tests.

3. **Formal definition of $D_{inter}$,$D_{intra}$**

(Reviewer 2vmL-W3, Reviewer NniG-W1.4)

**Response:**
 We added the formal definitions in the main text. This revision had already received positive feedback from Reviewer 2vmL before the incident.



## **Additional Revisions**

Beyond the common concerns above:

- Following Reviewer 2vmL, we added multi-run LLM experiments.
- Following Reviewer NniG, we added TSADM→LLM, LLM→TSADM, max, mean, and vote comparison baselines.
- We corrected theoretical issues by restating the probability space in the error accumulation analysis and showing it does not affect the final conclusion. (Reviewer NniG-W1.3)
- We provide more illustrations to some ambiguous part leading to misunderstandings:
  - The omitted $\log N$ term is a constant in the objective function and do not impact optimization process. Thus it can be omitted. (Reviewer NniG-W1.1)
  - The covariance term does not appear because the derivation involves the square of expectation, not the expectation of a squared term (Reviewer NniG-W1.3).
  - We reformat the proof of convergence claim to underline that the proposed loss function follows all the assumptions in the priori theorem (Reviewer NniG-W1.5)

---

### Meta-Review · Area_Chair_GVZe · 2026-01-04

**Summary:**

There are several concerns for this work cerntering around core assumptions of this work, derivations/definitions, and experimental settings. The authors did an effort to address these concerns, however, the concerns about validity of the assumptions are unclear if would satisfy the reviewers. Even if we assume some would increase their scores, which is possible due to the good effort from the authors, the overall score would remain in borderline era, hence, acceptance would be uncertain/unlikely. I hope the authors would find the comments useful and submit their work on an alternative venue.

**Reviewer Concerns:**

Outstanding concerns are mainly concentrated on various core definitions and assumptions, which are difficult to judge if the responses would have convinced the reviewers.

**Reviewer Scores:**

Even if we assume some reviewers would raise their scores, the overall score would be in borderline era, making it uncertain for acceptance

---

### Decision · Program_Chairs · 2026-01-26

Reject